# Scaffold-Conditioned Preference Triplets for Controllable Molecular Optimization with Large Language Models

## Abstract

Molecular property optimization is central to drug discovery, yet many deep learning methods rely on black-box scoring and offer limited control over scaffold preservation, often producing unstable or biologically implausible edits. While large language models (LLMs) are promising molecular generators, optimization remains constrained by the lack of chemistry-grounded preference supervision and principled data curation.

We introduce **Scaffold-Conditioned Preference Triplets (SCPT)**, a pipeline that constructs similarity-constrained triplets ⟨scaffold, better, worse⟩ via scaffold alignment and chemistry-driven filters for validity, synthesizability, and meaningful property gains. Using these preferences, we align a pretrained molecular LLM as a conditional editor, enabling property-improving edits that retain the scaffold.

Across single- and multi-objective benchmarks, SCPT achieves a stronger success–similarity trade-off than competitive baselines, especially under scaffold-constrained and multi-objective optimization settings. Compared with representative non-LLM molecular optimization methods, SCPT-trained LLMs are better suited to scaffold-constrained and multi-objective optimization. In addition, models trained on single-property and two-property supervision generalize effectively to three-property tasks, indicating promising extrapolative generalization under limited higher-order supervision. SCPT also provides controllable data-construction knobs that yield a predictable similarity–gain frontier, enabling systematic adaptation to diverse optimization regimes.

## 1 Introduction

In recent years, large language models (LLMs) have demonstrated strong sequence modeling capabilities for natural language generation, conditional editing, and preference alignment. Analogously, molecules can be viewed as a "molecular language": representations such as SMILES encode molecular structures as sequences, making "given a molecule, output an edited molecule" naturally cast as *constrained sequence editing* Weininger (1988); Krenn et al. (2020). This paradigm closely matches real-world drug and materials discovery, where chemists typically start from a validated lead compound and perform local, scaffold-conditioned modifications, aiming to improve target properties while maintaining key structural motifs and synthesizability. Such localized editing reduces exploration cost and improves experimental feasibility Xia (2017); Luukkonen et al. (2023); Zhang (2024); Xia et al. (2025).

In this work, a scaffold refers to the core structural framework of a molecule, typically including the main ring systems and linker structures around which peripheral substituents are modified. In lead optimization, preserving this scaffold is important because the core chemotype often carries key biological activity, synthetic feasibility, and chemical interpretability. Therefore, the goal is not simply to find any molecule with a higher oracle score, but to identify property-improving local edits that remain close to the original scaffold.

Despite its practical importance, *lead-based* molecular optimization is challenging not merely because it must produce valid molecules, but because it must make reasonable and controllable choices within a scaffold-local neighborhood Wang et al. (2025c). In realistic lead optimization, chemists rarely assume a single "correct" edit for a given input. Instead, they consider multiple chemically plausible candidates and rank them according to similarity constraints, synthesizability, and (single- or multi-objective) property gains Barshatski et al. (2021). The resulting supervision is inherently closer to preference ranking than to single-label

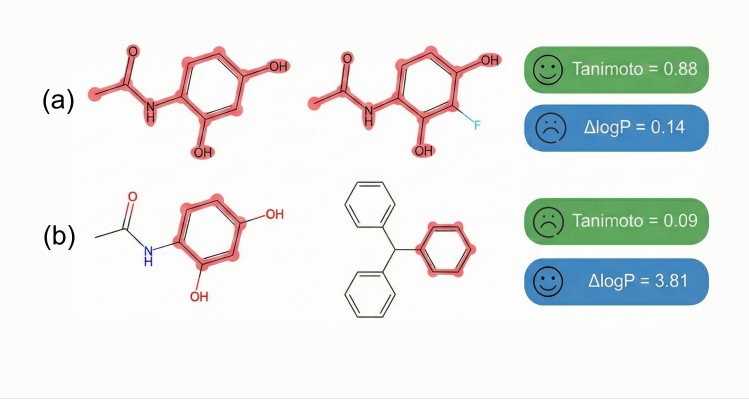

Figure 1: Optimization failure cases. (a) High similarity, negligible gain. (b) High gain, low similarity.

targets Thornber (1979); Meanwell (2011). However, most existing datasets and training signals rely on black-box scores or isolated pairs, and thus fail to explicitly encode which local edits are preferable under a given scaffold condition Luukkonen et al. (2023); Wang et al. (2022); Olivecrona et al. (2017). Figure 1 illustrates typical failure cases, and the experiments in Section 4.3 further corroborate this observation. This lack of scaffold-conditioned preference supervision hinders models from learning an interpretable, stable trade-off between locality and improvement, and makes it difficult to systematically study how data curation decisions shape optimization behavior.

We address this challenge through a *Scaffold-Conditioned Preference Triplets (SCPT) pipeline* for lead-based molecular optimization. The pipeline aligns and filters candidate molecules under scaffold/similarity constraints, and organizes them into Scaffold-Conditioned Preference Triplets ⟨context, better, worse⟩, so that relative quality within the same scaffold-local neighborhood is explicitly encoded. Importantly, the pipeline exposes controllable data curation knobs (e.g., similarity windows and the magnitude of property improvements), enabling a systematic characterization of the trade-off frontier between similarity preservation and property gains.

To validate the impact of our data construction choices, we treat a pretrained LLM as a conditional editor over molecular strings: pretraining provides strong priors over the syntax of molecular representations and common structural patterns Ouyang et al. (2022), while our constructed preference data injects optimization preferences about which scaffold-conditioned edits are more desirable. Building on supervised fine-tuning (SFT), we apply direct preference optimization (DPO) to bias generation toward edits preferred by the triplets Rafailov et al. (2023). This setup also serves as a unified and controlled experimental platform to quantify and explain how data construction choices systematically affect similarity, success rate, and property improvements.

Beyond the core evaluations of optimization performance, preference alignment, and data construction, we also broaden the empirical study in two directions that are important for practical deployment. We first benchmark the proposed LLM-based framework against representative non-LLM molecular optimization methods, including graph-based, VAE-based, and reinforcement-learning baselines. This comparison helps clarify whether the benefit of our approach lies specifically in better scaffold-preserving local editing, rather than in unconstrained exploration of chemical space. We then study a more data-limited setting in which training supervision is available only for single-property and two-property tasks, while evaluation is performed on unseen three-property combinations. This analysis is motivated by the fact that valid scaffold-conditioned supervision becomes increasingly scarce as more objectives are combined. Taken together, these additional evaluations extend the paper from method validation to a broader examination of robustness, fairness of comparison, and compositional generalization in molecular optimization.

- We propose a reproducible SCPT pipeline that systematizes chemistry heuristics into triplet-based supervision and provides tunable curation knobs for systematic analysis.

- Within a molecular string editing framework, we instantiate a pretrained LLM as a conditional editor and apply preference alignment to enable scaffold-conditioned property optimization.
- Through ablations and analysis, we reveal how data construction strategies govern the trade-off frontier between locality and gains, offering a controllable and interpretable paradigm for real-world lead-based molecular design.
- Our experiments show that, compared with representative non-LLM baselines, the proposed training paradigm enables molecular LLMs to operate more effectively under scaffold constraints, perform better in multi-objective optimization, and generalize more reliably to unseen higher-order property compositions.

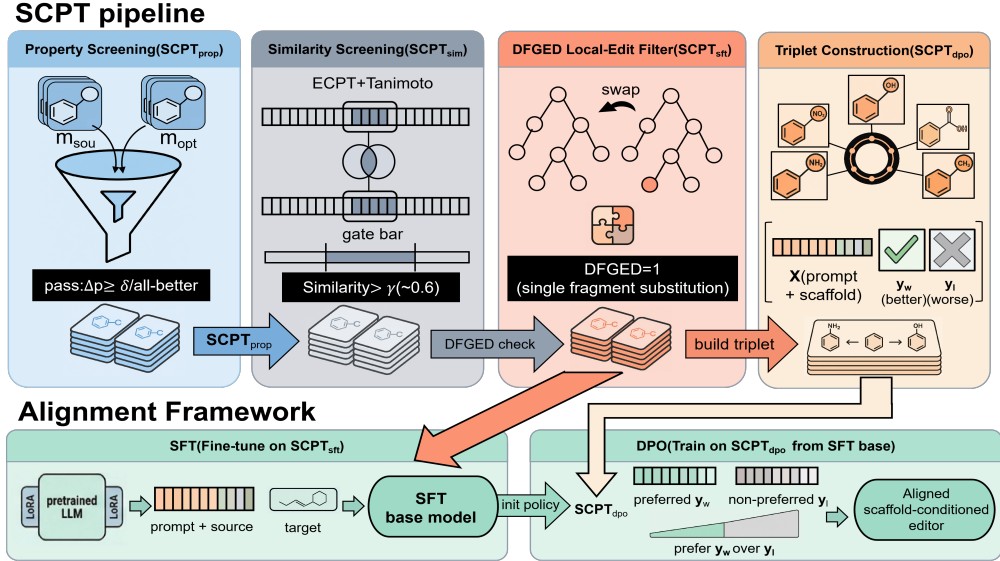

Figure 2: Overall two-stage framework of SCPT. (a) The SCPT data construction stage, which includes property-based filtering, similarity/local-edit filtering, and scaffold-conditioned triplet construction. (b) The alignment stage, which performs SFT on paired local edits and DPO on scaffold-conditioned preference triplets.

## 2 Related work

**Traditional molecular optimization models.** Graph-, sequence-, and 3D geometry-based models mainly differ in how they represent molecules. Graph approaches Reiser et al. (2022) such as JT-VAE Jin et al. (2018), F-RAG Lee et al. (2024), and GraphGA Jensen (2019) operate directly on molecular graphs via substructure assembly or fragment-level mutations. Sequence-based models treat molecules as SMILES or SELFIES strings Weininger (1988); Krenn et al. (2020); Transformer models such as Chemformer Irwin et al. (2022) and MolGPT Bagal et al. (2022) learn conditional generation and scaffold editing from linearized representations. 3D methods Powers et al. (2023), including GeoDiff Xu et al. (2022) and TFM-Flow Wang et al. (2025b), incorporate spatial coordinates to support conformational sampling and protein–ligand docking. While these models expand the reachable design space, they typically rely on engineered rewards or property predictors and offer limited control over scaffold preservation and synthesizability, often leading to *scaffold drift* during optimization.

**Large language models for molecular design.** Viewing chemistry as a molecular language, large-scale Transformers pretrained on SMILES or SELFIES strings have shown strong performance in molecular generation and property prediction Zhang et al. (2025); Wu et al. (2023), and SELFIES-based variants further guarantee chemical validity. LLMs have also been used as active optimizers or reasoning agents: MOLLEO Wang et al. (2025a) combines LLM-guided mutation with evolutionary search; Lico Nguyen &

Grover (2025) uses in-context surrogate models to reduce oracle calls; GellmoDey et al. (2025) proposed using the MuMOInstruct dataset to train a specialized LLM molecular optimization model, improving optimization performance; tool-augmented systems such as ChemCrow Bran et al. (2024) and DrugAssist integrate general-purpose LLMs with cheminformatics toolkits for interactive synthesis planning and optimization.

However, these approaches rarely optimize under explicit similarity or scaffold-conditioned constraints, and there is little structured, scaffold-conditioned preference data that encodes chemist-like modification preferences, leaving the alignment of LLMs for similarity-constrained molecular optimization largely unexplored.

## 3  Methodology

We propose a scaffold-conditioned preference alignment pipeline for lead-based molecular optimization. Molecules are represented as sequences in a formal molecular language, and the goal is to learn scaffold-conditioned policies that propose local, scaffold-preserving edits to a given lead molecule while improving clinically motivated molecular properties. As illustrated in Figure 2, the proposed methodology consists of two stages: (i) a scaffold-conditioned data construction pipeline that transforms static molecular data into preference-aligned supervision for local edits, followed by (ii) a model-agnostic alignment framework that can be instantiated with different molecular generative backbones. In this work, we adopt large language models as a standardized testbed to validate the effectiveness of the proposed pipeline.

### 3.1  Scaffold-Conditioned Molecular Optimization Problem Formulation

Let $\mathcal{M}$ denote the space of valid molecular strings. For a given source molecule $m_{\mathrm{sou}} \in \mathcal{M}$, we consider a collection of property predictors:

$$F_i : \mathcal{M} \to \mathbb{R}, \quad i = 1, \dots, N, \tag{1}$$

which quantify chemically or clinically relevant endpoints such as drug-likeness, target affinity, and ADMET (absorption, distribution, metabolism, excretion, and toxicity)–related properties. Our goal is to generate an optimized molecule $m_{\mathrm{opt}} \in \mathcal{M}$ that improves a weighted combination of these properties while preserving the scaffold of the source:

$$\max_{m_{\mathrm{opt}} \in \mathcal{M}} \sum_{i=1}^{N} w_i \, F_i(m_{\mathrm{opt}})$$
$$\text{s.t. } \mathrm{sim}(m_{\mathrm{sou}}, m_{\mathrm{opt}}) \geq \gamma. \tag{2}$$

where $w_i \geq 0$ reflect preference trade-offs, $\mathrm{sim}(\cdot, \cdot)$ measures structural similarity between two molecules, and $\gamma \in [0, 1]$ enforces scaffold preservation.

We instantiate this optimization via a conditional policy $\pi_\theta(m \mid x)$ implemented by an LLM. Given $x$, the model generates candidate sequences that correspond to edited molecules. Training shapes $\pi_\theta$ so that samples from $\pi_\theta(\cdot \mid x)$ align with the objective in Eq. 2. From the NLP perspective, this amounts to editing a structured language under similarity and preference constraints: the model must apply localized modifications that improve an implicit reward while remaining close to the original sequence.

### 3.2  Scaffold-Conditioned Preference Triplets Construction

We now describe how raw molecular data are transformed into structured preference signals. The pipeline consists of three components: property-based preference separation, similarity-constrained local-edit filtering, and SCPT construction.

**SCPT$_{\mathbf{prop}}$ and SCPT$_{\mathbf{sim}}$.**  Let $SCPT_{\mathrm{source}}$ denote the original labeled dataset, where each molecule $m$ is associated with a property vector:

$$\mathbf{p}(m) = \big(p_1(m), \dots, p_N(m)\big). \tag{3}$$

For any candidate pair $(m_{\text{sou}}, m_{\text{opt}})$, we define a directional property difference:

$$\Delta_p(m_{\text{sou}}, m_{\text{opt}}) = \sum_{i=1}^{N} w_i \, s_i \left( p_i(m_{\text{opt}}) - p_i(m_{\text{sou}}) \right),$$

where $\mathbf{s} \in \{-1, +1\}^N$ is a direction vector indicating whether each property should be increased $(+1)$ or decreased $(-1)$, and $\mathbf{w} \in \mathbb{R}_{\geq 0}^N$ are nonnegative weights denoting their relative importances. We retain only pairs with $\Delta_p(m_{\text{sou}}, m_{\text{opt}}) \geq \delta$, or those satisfying an "all-better" criterion in the multi-objective setting. The resulting set, denoted $SCPT_{\text{prop}}$, sharpens weak preference signals and substantially reduces the number of candidates that must undergo more expensive structural checks.

To enforce scaffold preservation and obtain interpretable local edits, we impose both fingerprint-level similarity and fragment-level minimal-edit constraints. For each pair $(m_{\text{sou}}, m_{\text{opt}}) \in SCPT_{\text{prop}}$, we first compute binary ECFP fingerprints $f_{\text{sou}}, f_{\text{opt}} \in \{0, 1\}^n$ and evaluate their Tanimoto similarity Cereto-Massagué et al. (2015):

$$\text{Tan}(m_{\text{sou}}, m_{\text{opt}}) = \frac{|f_{\text{sou}} \cap f_{\text{opt}}|}{|f_{\text{sou}}| + |f_{\text{opt}}| - |f_{\text{sou}} \cap f_{\text{opt}}|}. \tag{4}$$

Pairs with $\text{Tan}(\cdot, \cdot)$ below a threshold $\gamma$ are discarded, yielding a subset $SCPT_{\text{sim}}$.

**Single-fragment substitution.** Similarity alone does not guarantee simple local edits, so we additionally require that each pair corresponds to a *single-fragment substitution*. We measure fragment-level graph edit distance using a junction-tree representation. For each molecule $m$, we construct a junction tree Jin et al. (2018):

$$\text{JT}(m) = (V, E, \ell), \tag{5}$$

where nodes $V$ correspond to molecular fragments, edges $E$ encode fragment adjacency, and $\ell(v)$ is the SMILES label of fragment $v$ with attachment placeholders. The fragment-level distance between a pair $(m_{\text{sou}}, m_{\text{opt}})$ is defined as:

$$\text{DFGED}_{\text{frag}}(m_{\text{sou}}, m_{\text{opt}}) = \min_{\pi \in \Pi} \sum_{o \in \pi} c(o), \tag{6}$$

where $\Pi$ is the set of edit paths transforming $\text{JT}(m_{\text{sou}})$ into $\text{JT}(m_{\text{opt}})$, and $c(o)$ is the cost of each edit operation (unit cost is used for fragment insertion, deletion, and substitution). A pair is a single-fragment substitution if and only if

$$\text{DFGED}_{\text{frag}}(m_{\text{sou}}, m_{\text{opt}}) = 1 \tag{7}$$

and the optimal edit path contains exactly one fragment substitution with compatible attachment points and neighbors Chen et al. (2021).

**SCPT$_{\text{SFT}}$ and SCPT$_{\text{DPO}}$.** To further guarantee high atom-level overlap, we compute the maximum common subgraph (MCS) between $m_{\text{sou}}$ and $m_{\text{opt}}$ and enforce

$$\frac{|\text{MCS}(m_{\text{sou}}, m_{\text{opt}})|}{\max\left(|m_{\text{sou}}|, |m_{\text{opt}}|\right)} \geq \gamma_{\text{MCS}}, \tag{8}$$

with $\gamma_{\text{MCS}}$ set close to 0.9 in practice. Pairs that satisfy the property-margin, fingerprint similarity, single-fragment edit, and MCS constraints form the final paired dataset $SCPT_{\text{SFT}}$.

Preference-based training, such as DPO, requires triplets that specify which candidate is preferred under a given context. We construct scaffold-anchored triplets by grouping pairs from $SCPT_{\text{SFT}}$ according to their Bemis–Murcko scaffolds. For each scaffold, we collect molecules that share the same scaffold but differ by local substitutions, together with their properties. Within each scaffold group, we rank candidates according to $\Delta_p$ and create triplets $\langle scaffold, m_{\text{better}}, m_{\text{worse}} \rangle$ where $m_{\text{better}}$ has a higher directional gain than $m_{\text{worse}}$ under the same scaffold. We then convert each triplet into a scaffold-conditioned input/output tuple $\langle x, y_w, y_l \rangle$, where $x$ concatenates the textual prompt with the scaffold, and $y_w, y_l$ are the "better" and "worse" molecular sequences. The collection of all such triplets forms $SCPT_{\text{DPO}}$, which is the main training signal for preference alignment.

This construction differs from ordinary source–target pair supervision in that the preferred and dispreferred molecules are compared within the same scaffold-local neighborhood, making the supervision explicitly conditional on the retained molecular core rather than only on the final property score.

### 3.3 Aligning Molecular Language Models

Given $SCPT_{\text{SFT}}$ and $SCPT_{\text{DPO}}$, we align a pretrained molecular LLM in two stages. First, we train a base edit policy with SFT on $SCPT_{\text{SFT}}$. Second, we perform DPO on $SCPT_{\text{DPO}}$ to align the policy with chemist-like preferences over local edits.

We start from a pretrained LLM $\pi_\theta$ with chemistry-rich pretraining and apply low-rank adaptation (LoRA) rather than full parameter updates. For each pair $(m_{\text{sou}}, m_{\text{opt}}) \in SCPT_{\text{SFT}}$, we construct a text prompt $x$ that includes (i) a role description instructing the model to behave as a medicinal chemist, (ii) a description of the target property or task, and (iii) the SMILES/SELFIES string of the "worse" molecule $m_{\text{sou}}$. The target sequence $y$ is the string representation of the corresponding "better" molecule $m_{\text{opt}}$. SFT minimizes the standard next-token negative log-likelihood:

$$\mathcal{L}_{\text{SFT}}(\pi_\theta) = -\mathbb{E}_{(x,y)\sim D_{\text{SFT}}}\big[\log \pi_\theta(y \mid x)\big], \tag{9}$$

where $D_{\text{SFT}}$ is the empirical distribution over paired training examples.

To inject comparative structure and better reflect chemist-style decision making, we apply DPO on the scaffold-anchored triplets $SCPT_{\text{DPO}}$. Each training example consists of a scaffold-conditioned context $x$ and a preferred/non-preferred pair $(y_w, y_l)$, where $y_w$ and $y_l$ are the "better" and "worse" molecules under the same scaffold. DPO updates the policy using the log-probability gap between $y_w$ and $y_l$ as a sufficient statistic, via the objective

$$\mathcal{L}_{\text{DPO}}(\theta) = -\mathbb{E}_{(x,y_w,y_l)}\Big[\log \sigma\big(\beta \Delta_\theta(x, y_w, y_l)\big)\Big], \tag{10}$$

where $\sigma(\cdot)$ is the sigmoid function, $\beta > 0$ controls preference sharpness, and

$$\Delta_\theta(x, y_w, y_l) = \log \pi_\theta(y_w \mid x) - \log \pi_\theta(y_l \mid x) \tag{11}$$

is the log-probability difference between preferred and non-preferred molecules. Optimizing Eq. 10 shifts probability mass toward preferred edits while preserving the general behavior learned during SFT. Because the scaffold is explicitly part of the context $x$, the learned preferences are *scaffold-conditioned* and can later be analyzed to understand how data curation choices and DPO hyperparameters affect the trade-off between property gains and scaffold similarity.

## 4 Experiments

Our experiments are centered on a single message: the SCPT pipeline is the primary source of controllable, scaffold-conditioned optimization. We make this point by answering five tightly connected questions. First, what do the two key knobs (similarity control and the property-gap threshold) bring, and how do they shape the locality–gain trade-off that the model can feasibly learn? Second, does this curated data translate into consistent value at the model level, outperforming both prompting and chemistry-tuned baselines? Third, once the pipeline defines an explicit preference signal, can the model stably encode and learn that preference structure? Fourth, for scaffold-conditioned molecular optimization, are LLMs better suited than classical optimization-based generators? Fifth, when valid training data becomes increasingly sparse for higher-order objectives, can a LLM trained only on single-property and pairwise supervision extrapolate to unseen higher-order property compositions?

### 4.1 Experimental Setup

#### 4.1.1 Tasks and Datasets

**Single-property optimization.** We build on the MuMOInstruct corpus and reorganize molecular pairs into preference triplets of the form ⟨scaffold, better, worse⟩ for DPO training. We first consider eight stan-

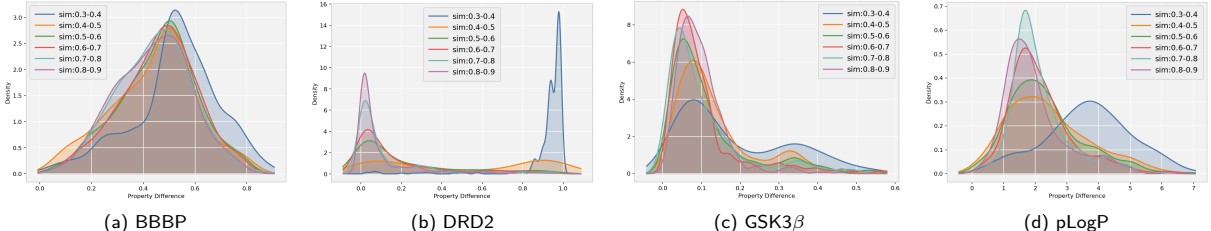

Figure 3: Density of Property Differences Between Source and Optimized Molecules Across Tasks under different similarity condition. The figure shows kernel density distributions of the property difference for each task.

Table 1: Single-property optimization results by similarity bin. Rows indicate similarity intervals and columns correspond to tasks. Full results appear in Table 12.

| Similarity | DRD2 | | | HIA | | | Mutag↓ | | | pLogP | | | GSK3$\beta$ | | |
|---|---|---|---|---|---|---|---|---|---|---|---|---|---|---|---|
| | SR | SIM | RI | SR | SIM | RI | SR | SIM | RI | SR | SIM | RI | SR | SIM | RI |
| 0.3–0.4 | 99.8 | 0.19 | 25.55 | 98.2 | 0.41 | 5.09 | 97.6 | 0.39 | 0.59 | 97.6 | 0.27 | 12.98 | 93.4 | 0.57 | 7.25 |
| 0.4–0.5 | 96.0 | 0.41 | 12.87 | 97.6 | 0.56 | 3.76 | 98.0 | 0.57 | 0.55 | 96.4 | 0.50 | 8.99 | 97.4 | 0.65 | 5.32 |
| 0.5–0.6 | 89.0 | 0.60 | 6.08 | 97.6 | 0.57 | 3.83 | 98.2 | 0.60 | 0.55 | 95.4 | 0.56 | 9.26 | 97.4 | 0.66 | 4.60 |
| 0.6–0.7 | 84.4 | 0.64 | 5.28 | 97.6 | 0.61 | 3.60 | 99.6 | 0.64 | 0.52 | 98.8 | 0.60 | 8.44 | 99.2 | 0.67 | 4.05 |
| 0.7–0.8 | 82.8 | 0.68 | 4.12 | 99.4 | 0.65 | 3.41 | 99.4 | 0.67 | 0.44 | 99.6 | 0.61 | 8.46 | 99.2 | 0.70 | 4.08 |
| 0.8–0.9 | 76.0 | 0.71 | 3.34 | 99.6 | 0.71 | 3.31 | 98.6 | 0.71 | 0.42 | 99.2 | 0.66 | 7.72 | 99.8 | 0.73 | 3.62 |

dard single-property tasks: pLogP, QED, BBBP, Mutag, HIA, DRD2, JNK3, GSK3$\beta$. The details of data construction are shown in the Appendix B.

**Multi-property optimization.** We also evaluate in more challenging multi-objective settings. There are five tasks: **BDP**: BBBP + DRD2 + pLogP; **BDQ**: BBBP + DRD2 + QED; **BPQ**: BBBP + pLogP + QED; **DPQ**: DRD2 + pLogP + QED; **BDPQ**: BBBP + DRD2 + pLogP + QED.

**Non-LLM baseline optimization.** For the non-LLM baselines, we adopt the task setup defined in the PMO framework Gao et al. (2022a). Specifically, the single-property experiments are conducted on four standard objectives: QED, pLogP, GSK3$\beta$, and DRD2. For the multi-property setting, we consider three-property combination tasks including **DPQ**: DRD2 + pLogP + QED; **GDP**: GSK3$\beta$ + DRD2 + pLogP; **GBD**: GSK3$\beta$ + QED + DRD2; **GQP**: GSK3$\beta$ + QED + pLogP.

### 4.1.2 Evaluation Metrics

We use three metrics to jointly assess optimization performance and structural fidelity:

**Success Rate (SR)** The fraction of test molecules for which at least one generated candidate satisfies all task-specific improvement thresholds.

**Similarity (SIM)** The Tanimoto similarity (on ECFP fingerprints) between the selected optimized molecule and its source, reflecting scaffold preservation and how local the edit is.

**Relative Improvement (RI)** The average relative change in the target properties between source and optimized molecules, aggregated across all properties in the task.

For each source molecule, the model generates 20 candidate molecules. We then score all valid candidates using the task-specific property oracle and select the best-scoring candidate as the final optimized molecule for computing SR, SIM, and RI. Thus, the reported metrics evaluate the best molecule found within a fixed generation budget rather than a single sampled output.

Formal definitions of all three metrics are given in Appendix C.1.

Table 2: Single-property optimization results by property-gap thresholds. Each row corresponds to a property-difference percentile range (the first row is the top 10% largest gaps), and each column to a task. Full results are reported in Table 13.

| Gap Range | DRD2 | | | HIA | | | Mutag↓ | | | pLogP | | | GSK3$\beta$ | | |
|---|---|---|---|---|---|---|---|---|---|---|---|---|---|---|---|
| | SR | SIM | RI | SR | SIM | RI | SR | SIM | RI | SR | SIM | RI | SR | SIM | RI |
| 0–0.1 | 76.0 | 0.67 | 3.88 | 99.4 | 0.64 | 3.75 | 98.8 | 0.68 | 0.45 | 99.8 | 0.60 | 8.97 | 99.6 | 0.70 | 5.17 |
| 0.1–0.2 | 76.4 | 0.68 | 4.12 | 99.4 | 0.66 | 3.37 | 99.6 | 0.66 | 0.49 | 99.4 | 0.63 | 7.69 | 98.0 | 0.72 | 2.71 |
| 0.2–0.3 | 79.6 | 0.69 | 3.30 | 99.0 | 0.66 | 3.55 | 99.8 | 0.67 | 0.48 | 99.8 | 0.64 | 6.57 | 98.6 | 0.73 | 2.74 |
| 0.3–0.4 | 75.8 | 0.70 | 2.92 | 99.4 | 0.66 | 3.36 | 99.6 | 0.67 | 0.48 | 99.8 | 0.64 | 6.46 | 97.4 | 0.72 | 2.20 |
| 0.4–0.5 | 75.8 | 0.70 | 2.65 | 99.4 | 0.66 | 2.97 | 99.6 | 0.67 | 0.50 | 99.8 | 0.66 | 5.74 | 97.2 | 0.72 | 1.55 |
| 0.5–0.6 | 74.2 | 0.71 | 2.52 | 99.6 | 0.66 | 2.61 | 100.0 | 0.66 | 0.51 | 99.4 | 0.68 | 4.98 | 96.8 | 0.73 | 1.65 |
| 0.6–0.7 | 88.2 | 0.68 | 2.03 | 99.6 | 0.66 | 2.72 | 100.0 | 0.68 | 0.47 | 99.6 | 0.67 | 4.89 | 96.0 | 0.71 | 1.54 |
| 0.7–0.8 | 80.2 | 0.71 | 1.72 | 99.8 | 0.68 | 2.19 | 99.6 | 0.66 | 0.55 | 99.8 | 0.67 | 5.31 | 95.8 | 0.70 | 1.63 |
| 0.8–0.9 | 80.2 | 0.72 | 1.50 | 99.4 | 0.69 | 2.07 | 100.0 | 0.67 | 0.50 | 99.4 | 0.68 | 4.69 | 96.8 | 0.73 | 1.30 |

### 4.1.3 Models and Baselines

We compare our method against both general-purpose and chemistry-focused language models: Closed-source general LLMs include ChatGPT-4o and Claude-3.5, open-source general LLMs cover LLaMA-3.1-8B-Instruct and Mistral-2.0-Instruct-7B, and chemistry-specialized models consist of ChemLLM and $LlaSmol_{Mistral}$.

General-purpose LLMs are evaluated in **zero-shot** and **five-shot** settings, where we prompt the model with natural language descriptions of the optimization task plus zero or five in-context examples. Chemistry-specific models are evaluated in their default (zero-shot) configuration.

We instantiate and train models with both Mistral-2.0-Instruct-7B and LLaMA-3.1-8B-Instruct as backbones, employing both SFT and DPO training strategies.

To further examine whether LLMs are well suited to scaffold-preserving molecular optimization, we additionally compare against representative non-LLM baselines from the PMO (Mol-Opt) framework, including REINVENT, GRAPHGA, and JTVAE.

We additionally include GENMOL as a diffusion-based molecular generation baseline. GenMol uses a diffusion-style generative process for molecular representations and has been evaluated on goal-directed generation and lead-optimization settings, making it a relevant recent non-LLM comparison for our scaffold-constrained optimization task.

Detailed comparisons of models and experimental settings are provided in the Appendix C.2 and C.3. Training-cost and hardware details are reported in Appendix C.3.2; all SCPT SFT and DPO runs use LoRA-based parameter-efficient fine-tuning and can be conducted on a single RTX 4090 GPU.

### 4.2 RQ1: Do the pipeline knobs behave as intended?

To isolate the effect of each pipeline knob, we vary only one factor at a time: across similarity-threshold settings, we keep the property-difference distribution matched, and across property-gap settings, we fix all data to the same similarity window. Detailed controlled-ablation protocols are provided in Appendix C.6.1 and C.6.2.

The first knob is similarity control. The point here is not to increase similarity, but to test whether explicitly conditioning on similarity produces a systematic, interpretable locality gain trade-off—so that locality becomes part of the training signal rather than an incidental byproduct of decoding. We train the model on SCPT$_{\text{sim}}$ constructed under different similarity-threshold settings for the experiment. Under this setup, a clear locality–gain tension emerges as shown in Table 1. DRD2 is the most illustrative: moving from a loose bucket (0.3–0.4) to a strict bucket (0.8–0.9) raises SIM from 0.19 to 0.71, but RI drops from 25.55 to 3.34 and SR falls from 99.8% to 76.0%. In other words, stringent similarity constraints directly compress the room for large improvements. Even when SR is near-saturated, similarity conditioning still imposes a clear cost on improvement: for pLogP, shifting from 0.3–0.4 to 0.8–0.9 increases SIM from 0.27 to 0.66 while RI

Table 3: Single-property results. Best values are bold, DPO results are highlighted in gray.

| Model | BBBP | | | DRD2 | | | HIA | | | Mutag | | | pLogP | | | QED | | | GSK3β | | | JNK3 | | |
|---|---|---|---|---|---|---|---|---|---|---|---|---|---|---|---|---|---|---|---|---|---|---|---|---|
| | SR↑ | SIM↑ | RI↑ | SR↑ | SIM↑ | RI↑ | SR↑ | SIM↑ | RI↑ | SR↑ | SIM↑ | RI↑ | SR↑ | SIM↑ | RI↑ | SR↑ | SIM↑ | RI↑ | SR↑ | SIM↑ | RI↑ | SR↑ | SIM↑ | RI↑ |
| LLaMA (0-shot) | 30.2 | **0.81** | 0.45 | 12.8 | 0.70 | 0.02 | 30.4 | 0.66 | 3.02 | 37.4 | **0.76** | 0.24 | 17.0 | 0.74 | 2.41 | 18.0 | 0.75 | 0.19 | 24.6 | 0.70 | 0.72 | 19.6 | 0.68 | 0.26 |
| LLaMA (5-shot) | 78.8 | 0.23 | 1.57 | 71.4 | 0.15 | **15.08** | 72.2 | 0.27 | 5.46 | 74.4 | 0.18 | 0.63 | 62.6 | 0.18 | 8.93 | 71.6 | 0.15 | **1.12** | 74.0 | 0.16 | 11.02 | 69.6 | 0.15 | **11.50** |
| Mistral (0-shot) | 30.4 | 0.72 | 0.51 | 17.6 | 0.72 | 0.72 | 57.2 | **0.72** | 1.28 | 41.0 | 0.69 | 0.33 | 32.2 | 0.74 | 2.71 | 12.8 | 0.67 | 0.21 | 29.0 | 0.67 | 0.71 | 23.8 | 0.64 | 0.27 |
| Mistral (5-shot) | 70.6 | 0.67 | 0.93 | 28.6 | 0.64 | 1.92 | 72.2 | 0.68 | 1.88 | 59.8 | 0.63 | 0.39 | 64.0 | 0.70 | 3.27 | 50.0 | 0.68 | 0.27 | 50.6 | 0.69 | 1.40 | 49.6 | 0.67 | 1.58 |
| GPT-4o (0-shot) | 56.0 | 0.70 | 0.66 | 19.0 | 0.73 | 0.25 | 58.4 | 0.71 | 1.40 | 44.0 | 0.70 | 0.27 | 43.2 | 0.72 | 3.20 | 38.4 | 0.75 | 0.24 | 41.8 | 0.68 | 0.91 | 30.4 | 0.67 | 0.27 |
| GPT-4o (5-shot) | 64.8 | 0.64 | 0.80 | 33.4 | 0.62 | 1.93 | 74.0 | 0.61 | 2.74 | 58.4 | 0.64 | 0.34 | 64.2 | 0.64 | 4.07 | 59.6 | 0.67 | 0.31 | 51.0 | 0.61 | 2.47 | 46.8 | 0.64 | 0.98 |
| Claude-3.5 (0-shot) | 17.6 | 0.72 | 0.58 | 15.4 | 0.73 | 0.79 | 28.8 | 0.63 | 2.33 | 26.4 | 0.66 | 0.29 | 37.4 | 0.73 | 1.54 | 19.4 | 0.71 | 0.21 | 27.2 | 0.69 | 0.77 | 26.4 | 0.70 | 0.18 |
| Claude-3.5 (5-shot) | 47.4 | 0.72 | 0.62 | 28.2 | 0.69 | 1.63 | 52.2 | 0.63 | 1.62 | 61.4 | 0.68 | 0.34 | 56.6 | 0.69 | 2.35 | 45.8 | 0.71 | 0.31 | 50.4 | 0.65 | 2.00 | 42.2 | 0.64 | 1.19 |
| ChemLLM | 33.0 | 0.79 | 0.46 | 12.8 | **0.80** | 0.01 | 23.6 | 0.74 | 1.95 | 27.4 | 0.73 | 0.27 | 14.4 | **0.78** | 4.93 | 7.4 | **0.77** | 0.20 | 20.0 | **0.77** | 0.88 | 20.6 | **0.75** | 0.57 |
| $LlaSmol_{Mistral}$ | 79.6 | 0.25 | 1.53 | 63.6 | 0.30 | 0.18 | 70.2 | 0.50 | 2.19 | 63.0 | 0.28 | 0.59 | 44.6 | 0.45 | 3.75 | 41.6 | 0.30 | 0.49 | 69.6 | 0.33 | 1.61 | 64.4 | 0.35 | 0.50 |
| $SFT\text{-}LLM_{Mistral}$ | 99.55 | 0.56 | 1.76 | **98.10** | 0.57 | 8.66 | 98.95 | 0.56 | 5.05 | **99.80** | 0.57 | 0.68 | **99.95** | 0.57 | 9.81 | **100.00** | 0.57 | 0.81 | **98.80** | 0.59 | 10.26 | 98.20 | 0.58 | 3.94 |
| $DPO\text{-}LLM_{Mistral}$ | 99.53 | 0.52 | 1.97 | 95.60 | 0.55 | 14.82 | 97.33 | 0.49 | 6.74 | 99.40 | 0.52 | **0.85** | 99.60 | 0.51 | 17.43 | 99.53 | 0.53 | 0.92 | 98.25 | 0.52 | 17.12 | **98.90** | 0.49 | 8.59 |
| $SFT\text{-}LLM_{LLaMA}$ | **99.73** | 0.58 | 1.76 | 96.60 | **0.58** | 7.76 | 98.47 | 0.56 | 5.69 | 99.53 | 0.56 | 0.67 | 99.93 | 0.57 | 9.02 | 99.80 | 0.57 | 0.86 | 98.60 | 0.51 | 12.79 | 98.60 | 0.55 | 5.77 |
| $DPO\text{-}LLM_{LLaMA}$ | 99.53 | 0.51 | **1.99** | 97.80 | 0.55 | 10.63 | **99.47** | 0.52 | **7.46** | 99.00 | 0.52 | 0.78 | 99.47 | 0.48 | **18.10** | 99.80 | 0.54 | 0.98 | 98.00 | 0.49 | **20.01** | 98.20 | 0.47 | 7.81 |

decreases from 12.98 to 7.72. This is precisely the behavior we want the pipeline to make explicit—similarity acts as a local-edit prior that constrains how preference signals can be expressed. Across tasks, we also see diminishing returns: pushing similarity beyond a moderate window yields only marginal SIM gains while RI continues to decline, which is why a mid-range similarity window is a sensible operating point rather than the higher the better. Consistent with the table-level results, Figure 3 visualizes the corresponding shifts in the property-difference distributions under different similarity constraints.

The second knob is the property-gap threshold. If the required margin is too small, the boundary between better and worse becomes fuzzy, and preference learning can degenerate into merely satisfying minimal constraints; if it is too large, the data become sparse and skewed toward rare (and potentially idiosyncratic) transformations. For this experiment, we train the model on $SCPT_{prop}$ under different property-threshold settings. In our ablations, increasing the property gap typically *reduces* achievable RI while leaving SIM relatively stable, consistent with a coverage–sharpness trade-off, as shown in Table 2. For example, under DRD2, increasing the margin from 0–0.1 to 0.3–0.4 changes RI from 3.88 to 2.92 (with SIM remaining around 0.68–0.69). This supports viewing $\delta$ as a data-quality dial rather than a simple filter: it governs how much headroom the model has to learn meaningful preferences under scaffold constraints.

Practical takeaway. These results provide a simple guideline for choosing the operating point of SCPT. When the application prioritizes conservative lead refinement, scaffold retention, or downstream SAR analysis, stricter similarity constraints should be preferred, with the expected cost of smaller property gains. In contrast, when the goal is exploratory optimization or hit expansion, looser similarity constraints can provide larger relative improvements, but may allow more substantial structural changes. Therefore, the similarity threshold should not be viewed as a universally monotonic objective; rather, it controls the desired position on the similarity–improvement frontier.

### 4.3 RQ2: Can a preference-aligned LLM editor achieve meaningful gains without leaving the scaffold neighborhood?

The baselines in this section instantiate different algorithmic paradigms. Prompt-only LLMs test whether natural-language instructions alone can induce scaffold-preserving optimization. Chemistry-specialized LLMs test whether molecular pretraining is sufficient without task-specific preference supervision. SCPT-SFT tests whether paired local-edit supervision can teach feasible scaffold-conditioned edits, while SCPT-DPO further tests whether within-scaffold preferences can bias the model toward higher-gain edits. Therefore, the comparison is not only a performance benchmark, but also an algorithmic decomposition of where the benefit of SCPT comes from.

Guided by the controlled analysis in RQ1, we adopt a practical operating point for the subsequent experiments (with detailed settings provided in the Appendix C.3) and move from data-level analysis to model-level evaluation. Under this setup, RQ2 asks whether the scaffold-conditioned preferences constructed by SCPT can be translated into a more effective editing policy than baseline LLMs for molecular optimization under scaffold constraints. Viewed from this perspective, the pattern is strikingly consistent: prompted general-purpose LLMs typically face an awkward trade-off, either maintaining high similarity while failing to satisfy property constraints reliably, or achieving larger improvements by drifting away from the original scaffold. In single-property DRD2, as shown in Table 3, for instance, LLaMA (0-shot) attains SR 12.8 with SIM 0.70

Table 4: Multi-property results. Best values are bold, DPO results are highlighted in gray.

| Model | BDP | | | BDQ | | | BPQ | | | DPQ | | | BDPQ | | |
|---|---|---|---|---|---|---|---|---|---|---|---|---|---|---|---|
| | SR ↑ | SIM ↑ | RI ↑ | SR ↑ | SIM ↑ | RI ↑ | SR ↑ | SIM ↑ | RI ↑ | SR ↑ | SIM ↑ | RI ↑ | SR ↑ | SIM ↑ | RI ↑ |
| Mistral (0-shot) | 6.6 | **0.81** | 0.68 | 3.0 | **0.76** | 0.53 | 15.8 | 0.73 | 0.51 | 2.2 | **0.65** | 0.41 | 3.2 | 0.77 | 0.87 |
| LLaMA (0-shot) | 22.0 | 0.73 | 0.74 | 2.2 | 0.64 | 0.53 | 28.4 | 0.64 | 0.72 | 2.6 | 0.62 | 0.32 | 5.2 | **0.80** | 0.62 |
| Claude-3.5 (0-shot) | 19.6 | 0.66 | 1.05 | 13.0 | 0.62 | 1.14 | 56.0 | 0.62 | 0.86 | 11.0 | 0.54 | 0.51 | 8.0 | 0.60 | 1.34 |
| GPT-4o (0-shot) | 7.8 | 0.69 | 0.90 | 2.0 | 0.69 | 0.62 | 36.4 | 0.73 | 0.42 | 2.8 | 0.57 | 0.50 | 1.8 | 0.71 | 0.39 |
| Mistral (5-shot) | 35.2 | 0.64 | 2.10 | 17.0 | 0.60 | 2.32 | 68.6 | 0.63 | 0.79 | 10.4 | 0.54 | 1.10 | 11.0 | 0.69 | 0.96 |
| LLaMA (5-shot) | 35.4 | 0.57 | 2.71 | 16.6 | 0.43 | 5.70 | 34.6 | 0.70 | 0.64 | 8.2 | 0.44 | 3.02 | 9.6 | 0.54 | 3.45 |
| Claude-3.5 (5-shot) | 35.4 | 0.50 | 2.43 | 29.4 | 0.43 | 3.80 | 76.8 | 0.53 | 1.24 | 29.2 | 0.37 | 2.87 | 20.8 | 0.35 | 3.53 |
| GPT-4o (1-shot) | 9.4 | 0.69 | 0.79 | 7.6 | 0.66 | 0.61 | 40.0 | **0.75** | 0.41 | 7.0 | 0.62 | 0.44 | 3.4 | 0.70 | 0.61 |
| ChemLLM | 0.2 | 0.17 | 1.20 | 1.0 | 0.55 | 0.82 | 4.8 | 0.29 | 0.96 | 0.6 | 0.28 | 0.42 | 0.0 | 0.00 | 0.00 |
| LlaSmol$_{Mistral}$ | 43.6 | 0.62 | 1.09 | 31.4 | 0.66 | 0.93 | 86.0 | 0.58 | 0.84 | 24.0 | 0.57 | 0.61 | 14.0 | 0.62 | 1.03 |
| SFT-LLM$_{Mistral}$ | 81.6 | 0.56 | 3.87 | 83.95 | 0.55 | 4.98 | 97.05 | 0.54 | 1.45 | 62.85 | 0.51 | 2.42 | 57.0 | 0.52 | 3.05 |
| DPO-LLM$_{Mistral}$ | 81.93 | 0.50 | 5.15 | 85.87 | 0.52 | 6.22 | 97.07 | 0.47 | 1.74 | **70.13** | 0.44 | **4.30** | 65.13 | 0.43 | 5.65 |
| SFT-LLM$_{LLaMA}$ | 78.6 | 0.56 | 3.64 | 78.95 | 0.56 | 4.49 | 94.55 | 0.50 | 1.55 | 57.6 | 0.51 | 2.12 | 46.15 | 0.50 | 3.36 |
| DPO-LLM$_{LLaMA}$ | **84.07** | 0.47 | **6.67** | **86.67** | 0.51 | **6.52** | **97.13** | 0.45 | **1.98** | 68.33 | 0.46 | 3.39 | 59.2 | 0.42 | **5.97** |

but essentially no improvement (RI 0.02). LLaMA (5-shot) raises SR to 71.4 and RI to 15.08, but SIM collapses to 0.15—success is obtained by leaving the scaffold condition.

Chemistry-specialized LLMs might seem, at first glance, as if they should resolve this tension, yet the results reveal limitations that actually strengthen the motivation for our pipeline. ChemLLM tends to be conservative (high SIM) but struggles to optimize consistently: on DRD2 it reaches SR 12.8 with SIM 0.80, yet RI is only 0.01. Put differently, chemistry knowledge alone does not automatically translate into a reliable editing policy under task-specific thresholds. $LLaSmol_{Mistral}$ sits at the opposite extreme. It is effective at producing molecules that 'pass the test' (e.g., BBBP SR 79.6%), but often does so with substantial scaffold drift (BBBP SIM 0.25; DRD2 SIM 0.30), and gains can still be limited on certain endpoints (e.g., DRD2 RI 0.18). It behaves more like a broadly capable chemistry assistant than a scaffold-preserving optimizer.

In contrast, with SCPT data curated by our pipeline, model behavior changes sharply: SCPT$_{SFT}$ already teaches the backbone feasible local edits (SR near saturation on most single-property tasks), and SCPT$_{DPO}$ further biases the model toward *higher-gain* edits without exiting the scaffold-conditioned regime. This contrast becomes even more pronounced in multi-property optimization, where the interaction of multiple constraints increases the tendency toward scaffold deviation in the absence of explicit control. In this regime, both general LLMs and chemistry-specialized LLMs are fragile: as shown in Table 4, ChemLLM essentially collapses on the hardest four-property task BDPQ (SR 0.0), and even $LLaSmol_{Mistral}$ reaches only SR 14.0 on BDPQ (SIM 0.62, RI 1.03). By comparison, once trained on SCPT, the same backbone becomes far closer to a usable optimizer, as DPO shifts probability mass toward locally preferred edits under the same scaffold context: it improves BDPQ to SR 65.13 / RI 5.65 (Mistral backbone) and SR 59.2 / RI 5.97 (LLaMA backbone), with SIM around 0.39 and 0.38 respectively—lower than SFT, but still within a controlled range rather than unconstrained drift.

For the SCPT-based SFT and DPO training stages, we performed five independent runs using different random seeds to mitigate the randomness introduced by a single run and to evaluate result stability. Table 3 and Table 4 present the averaged results over the repeated runs, while Table 14, Table 15, Table 16, Table 17 additionally report the mean and standard deviation. The relatively small standard deviations across metrics indicate that the observed improvements are stable under different random seeds, thereby providing further evidence for the robustness and reliability of our findings.

## 4.4 RQ3: Is preference alignment controllable at the model level, such that DPO can systematically tune the scaffold–gain trade-off?

The third question is whether the pipeline truly yields a usable preference signal—one that DPO can exploit robustly, rather than behaving like a brittle trick. We probe this by a small sweep over DPO learning rate ($lr$) and $\beta$, and observe a consistent frontier: aggressive updates (higher $lr$, smaller $\beta$) increase SR/RI but reduce SIM; conservative updates recover similarity but weaken gains.

Table 5: DPO results under different hyperparameter settings. Rows correspond to hyperparameter combinations; columns correspond to tasks.

| Param | BDP | | | BDPQ | | | BDQ | | | BPQ | | | DPQ | | |
|---|---|---|---|---|---|---|---|---|---|---|---|---|---|---|---|
| | RI↑ | SR↑ | SIM↑ | RI↑ | SR↑ | SIM↑ | RI↑ | SR↑ | SIM↑ | RI↑ | SR↑ | SIM↑ | RI↑ | SR↑ | SIM↑ |
| lr1e-05+beta0.1 | 7.92 | 92.2 | 0.33 | 8.32 | 82.0 | 0.27 | 8.60 | 89.6 | 0.41 | 2.17 | 96.8 | 0.34 | 7.29 | 83.2 | 0.30 |
| lr1e-05+beta0.3 | 6.46 | 85.4 | 0.40 | 7.02 | 72.6 | 0.32 | 6.99 | 88.2 | 0.47 | 1.98 | 97.4 | 0.40 | 5.49 | 77.2 | 0.36 |
| lr1e-05+beta0.5 | 6.40 | 84.8 | 0.42 | 6.63 | 72.2 | 0.34 | 6.82 | 86.8 | 0.48 | 1.93 | 97.8 | 0.41 | 4.93 | 76.8 | 0.38 |
| lr5e-06+beta0.1 | 6.11 | 85.6 | 0.42 | 6.42 | 74.4 | 0.35 | 7.17 | 88.0 | 0.46 | 1.93 | 97.2 | 0.41 | 5.49 | 77.6 | 0.37 |
| lr5e-06+beta0.3 | 5.48 | 83.2 | 0.46 | 5.85 | 66.4 | 0.39 | 6.16 | 84.8 | 0.51 | 1.78 | 96.8 | 0.44 | 4.25 | 72.2 | 0.41 |
| lr5e-06+beta0.5 | 5.12 | 81.6 | 0.48 | 5.45 | 64.6 | 0.39 | 5.97 | 85.4 | 0.51 | 1.74 | 96.8 | 0.46 | 3.85 | 71.4 | 0.43 |
| lr1e-06+beta0.1 | 4.10 | 79.0 | 0.53 | 3.89 | 55.2 | 0.47 | 5.29 | 82.4 | 0.54 | 1.58 | 95.8 | 0.51 | 2.76 | 62.8 | 0.49 |
| lr1e-06+beta0.3 | 4.05 | 78.2 | 0.54 | 3.75 | 54.4 | 0.48 | 5.24 | 81.8 | 0.55 | 1.57 | 96.0 | 0.51 | 2.51 | 61.4 | 0.50 |
| lr1e-06+beta0.5 | 4.01 | 77.0 | 0.54 | 3.67 | 53.6 | 0.48 | 5.24 | 81.4 | 0.55 | 1.57 | 96.0 | 0.51 | 2.74 | 60.6 | 0.50 |
| lr5e-07+beta0.1 | 3.94 | 78.6 | 0.55 | 3.63 | 53.2 | 0.49 | 5.17 | 81.8 | 0.55 | 1.53 | 95.0 | 0.52 | 2.55 | 61.4 | 0.51 |
| lr5e-07+beta0.3 | 3.97 | 78.4 | 0.55 | 3.49 | 52.6 | 0.49 | 5.16 | 81.0 | 0.55 | 1.52 | 96.2 | 0.52 | 2.62 | 60.8 | 0.51 |
| lr5e-07+beta0.5 | 3.83 | 78.4 | 0.55 | 3.58 | 51.2 | 0.49 | 5.13 | 81.6 | 0.55 | 1.52 | 95.8 | 0.52 | 2.46 | 61.2 | 0.50 |
| SFT | 3.46 | 78.4 | 0.56 | 3.27 | 51.8 | 0.52 | 4.99 | 79.8 | 0.56 | 1.44 | 95.4 | 0.54 | 2.35 | 59.6 | 0.52 |

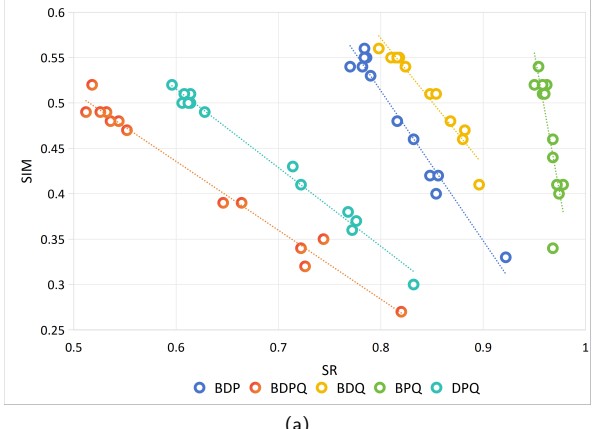
(a)

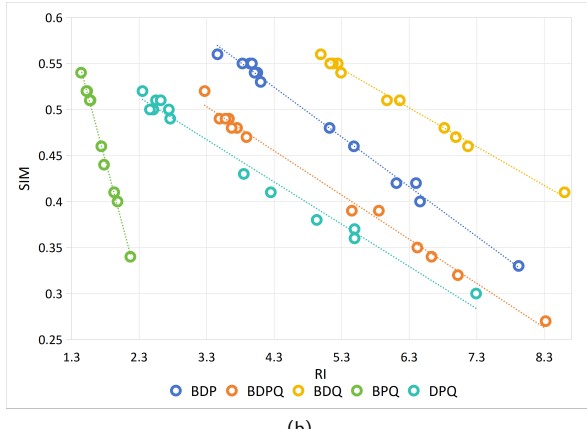
(b)

Figure 4: Visualization of DPO hyperparameter trends. By plotting the data points from Table 5 and fitting a curve, we observe a clear negative correlation. (a) SIM vs. SR; (b) SIM vs. RI.

As shown in Table 5, on the hardest task BDPQ, a high-update configuration ($lr = 1 \times 10^{-5}, \beta = 0.1$) reaches SR 82.0 and RI 8.32 at SIM 0.27, whereas a much smaller learning rate ($lr = 1 \times 10^{-6}, \beta = 0.1$) yields more conservative edits but substantially lower SR and RI . Holding $lr$ fixed, increasing $\beta$ similarly pushes the model toward safer edits: at $lr = 1 \times 10^{-5}$, raising $\beta$ from 0.1 to 0.5 increases SIM (0.27→0.34) while SR decreases (82.0→72.2) and RI decreases (8.32→6.63). We further visualize this trend in Figure 4; notably, each task exhibits the same consistent shift in behavior under these hyperparameter changes.

This predictability is crucial for our core argument. It suggests that once the data pipeline converts chemical heuristics into scaffold-conditioned preference triplets, DPO is not merely overfitting a training format; instead, it exposes a **tunable preference-alignment mechanism**. By turning this knob, we can meet different scaffold-preservation requirements while maintaining meaningful improvements.

### 4.5 RQ4: Why use LLMs instead of traditional molecular optimization models?

A natural question is whether SCPT-trained LLMs are genuinely preferable to standard molecular optimization paradigms under scaffold-conditioned local editing. Classical graph-, VAE-, and RL-based optimizers typically search chemical space using scalarized property rewards, sometimes with a similarity term added to the objective. We additionally include GenMol as a diffusion-based molecular generator. Since our setting

Table 6: Comparison with representative non-LLM baselines under a similarity-aware PMO objective. The main comparison uses a generation budget of 20.

| Method | DRD2 | | | GSK3$\beta$ | | | pLogP | | | QED | | |
|---|---|---|---|---|---|---|---|---|---|---|---|---|
| | SR↑ | SIM↑ | RI↑ | SR↑ | SIM↑ | RI↑ | SR↑ | SIM↑ | RI↑ | SR↑ | SIM↑ | RI↑ |
| GraphGA (20) | 30.80 | 0.210 | 12.86 | **100.00** | 0.077 | 15.48 | **100.00** | 0.154 | 11.00 | **100.00** | 0.159 | 1.20 |
| JTVAE (20) | **99.60** | 0.111 | **477.19** | 84.60 | 0.122 | 5.86 | **100.00** | 0.124 | 15.79 | 99.80 | 0.151 | 1.16 |
| REINVENT (20) | 52.00 | 0.188 | 38.41 | 57.59 | 0.165 | 0.81 | **100.00** | 0.105 | **19.06** | **100.00** | 0.138 | **1.32** |
| GenMol (20) | 10.80 | 0.448 | 1.54 | 53.35 | 0.449 | 2.99 | 58.00 | 0.442 | 4.16 | 62.80 | 0.427 | 0.75 |
| SFT-Mistral | 98.10 | 0.570 | 8.65 | 98.80 | **0.590** | 10.26 | 99.95 | **0.570** | 9.81 | **100.00** | **0.570** | 0.81 |
| DPO-Mistral | 95.60 | 0.547 | 14.82 | 98.25 | 0.520 | 17.12 | 99.60 | 0.510 | 17.43 | 99.53 | 0.533 | 0.92 |
| SFT-LLaMA | 96.60 | **0.583** | 7.76 | 98.60 | 0.510 | 12.79 | 99.93 | **0.570** | 9.02 | 99.80 | **0.570** | 0.86 |
| DPO-LLaMA | 97.80 | 0.550 | 10.63 | 98.00 | 0.490 | **20.01** | 99.47 | 0.480 | 18.10 | 99.80 | 0.543 | 0.98 |

is source-conditioned scaffold-preserving local editing, GenMol is used as a executable diffusion comparison rather than as a fully retrained scaffold-conditioned editor. To align the non-LLM baselines with our task formulation, we define their fitness as the sum of source-molecule similarity and the task-specific property objective. The main comparison uses a matched generation budget of 20, which is the same budget used by our LLM-based models.

Table 6 reveals a clear pattern: the main advantage of LLMs is not unrestricted property maximization, but high-quality local optimization under strong structural preservation. Across all four tasks (DRD2, GSK3$\beta$, pLogP, and QED), the LLM-based models consistently achieve much higher similarity to the source molecule, with SIM typically in the range of 0.48–0.59, whereas earlier non-LLM baselines remain in a very low-similarity regime, while GenMol operates in a noticeably higher-similarity regime but still lags behind SCPT-trained LLMs in SR and RI on most tasks.

At the same time, the LLM models maintain near-saturated success rates while preserving substantially higher similarity. This is particularly important for lead optimization, where the goal is not merely to find molecules with improved properties, but to identify plausible *local edits* around a reference scaffold. From this perspective, the comparison is informative: some non-LLM methods can indeed achieve extremely high RI on tasks such as DRD2 or pLogP, but these gains are usually accompanied by very low similarity, indicating that the improvement is obtained by moving far away from the starting molecule rather than refining it locally. In contrast, our models operate in a much more conservative and scaffold-preserving regime. Notably, on GSK3$\beta$, DPO-LLaMA achieves 20.01 RI with 0.49 similarity and 98.0 SR, outperforming all non-LLM baselines not only in similarity but also in RI, showing that strong structural preservation does not necessarily imply weak optimization.

The larger-budget results in Table 18 further support this interpretation. As the search budget for non-LLM baselines increases from 20 to 50, 100, and 200, RI often continues to rise dramatically, but similarity does not improve accordingly and often remains very low. This suggests that additional search budget is primarily spent exploring molecules that drift farther away from the source structure, rather than discovering better local scaffold-preserving edits. Overall, these results suggest that LLMs are especially well suited to molecular optimization when the task is formulated as source-conditioned local editing around a reference scaffold, rather than unconstrained search in chemical space.

## 4.6 RQ5: Can low-order supervision compensate for the scarcity of higher-order multi-property training data?

During SCPT data construction, we observed that valid training data becomes substantially sparser as the number of jointly optimized properties increases. This is expected: under our setting, a candidate pair must simultaneously satisfy multiple directional property constraints while also remaining within a scaffold-conditioned local neighborhood defined by similarity and structural filters. As more properties are combined, the feasible candidate space shrinks rapidly, resulting in far fewer valid preference pairs and weaker supervision for direct training on high-order objectives.

Table 7: Compositional generalization to unseen three-property optimization tasks. LLM models are trained only on single-property and pairwise-property optimization data. Non-LLM baselines are evaluated under the same PMO setting with a similarity-aware fitness, and the main comparison uses a matched generation budget of 20.

| Method | DRD2+pLogP+QED | | | GSK3$\beta$+DRD2+pLogP | | | GSK3$\beta$+QED+DRD2 | | | GSK3$\beta$+QED+pLogP | | |
|---|---|---|---|---|---|---|---|---|---|---|---|---|
| | SR↑ | SIM↑ | RI↑ | SR↑ | SIM↑ | RI↑ | SR↑ | SIM↑ | RI↑ | SR↑ | SIM↑ | RI↑ |
| GraphGA (20) | 9.0 | 0.154 | 3.13 | 19.9 | 0.144 | 11.04 | 5.9 | 0.101 | 6.30 | 4.1 | 0.153 | 0.60 |
| JTVAE (20) | 51.0 | 0.101 | 2.44 | 69.3 | 0.115 | 8.02 | 30.7 | 0.101 | 13.82 | 91.8 | 0.094 | 1.79 |
| REINVENT (20) | 35.8 | 0.112 | 2.05 | 4.8 | 0.137 | 7.86 | 0.8 | 0.135 | 1.47 | 85.4 | 0.109 | 1.69 |
| SFT-LLaMA | 57.2 | **0.530** | 2.17 | 70.4 | **0.610** | 17.55 | 67.6 | **0.620** | 9.51 | 85.0 | **0.580** | 1.71 |
| DPO-LLaMA | 68.0 | 0.420 | **5.74** | **82.8** | 0.520 | 51.67 | 72.4 | 0.530 | 21.45 | **92.6** | 0.500 | **3.35** |
| SFT-Mistral | 60.8 | 0.520 | 2.28 | 69.0 | 0.580 | 32.16 | 67.6 | 0.580 | 10.94 | 85.4 | 0.560 | 1.56 |
| DPO-Mistral | **70.6** | 0.460 | 4.61 | 79.8 | 0.490 | **59.86** | **75.2** | 0.510 | **24.13** | 90.8 | 0.530 | 3.01 |

This motivates an important question: *can the model learn reusable optimization behaviors from lower-order objectives and extrapolate them to unseen higher-order compositions?* To study this, we train the LLMs using only single-property and pairwise-property optimization data, and evaluate them on four unseen three-property objectives. This setting directly probes *compositional extrapolation*: rather than memorizing specific task configurations, the model must recombine optimization skills acquired from lower-order supervision and transfer them to higher-order compositions that never appear during training. It also allows us to examine whether our preference-alignment framework, especially DPO, improves this extrapolative generalization ability. Detailed experimental setups are provided in the AppendixC.3.3.

Table 7 shows that the LLM-based models generalize successfully to all unseen three-property tasks. Even without any three-property supervision, they maintain strong optimization performance across all four objectives, with success rates ranging from 57.2 to 92.6, similarity ranging from 0.42 to 0.62. This indicates that the models are not merely fitting the seen single- and two-property tasks independently; instead, they learn source-conditioned editing behaviors that can be meaningfully recomposed at test time.

More importantly, the LLM models substantially outperform representative non-LLM baselines under the same PMO setting. At the matched budget of 20, all non-LLM methods remain in a low-similarity regime (roughly 0.09–0.15), whereas the LLM models consistently operate in a much more scaffold-conditioned region (0.42–0.62). On a per-task basis, the best LLM similarity is approximately 3.4× to 4.6× higher than the best non-LLM counterpart. Importantly, this advantage in similarity does not come at the expense of optimization quality: the best LLM model also achieves the highest RI on all four tasks, with especially large margins on GSK3$\beta$+DRD2+pLogP (59.86 vs. 11.04) and GSK3$\beta$+QED+DRD2 (24.13 vs. 13.82). Likewise, the best LLM model attains the highest SR on all four tasks. These results suggest that LLMs are better suited for higher-order molecular optimization when the task is framed as source-conditioned local editing around a reference scaffold. This pattern also suggests that, in multi-objective optimization, classical non-LLM methods often struggle to balance competing objectives under scaffold constraints: they may improve the scalarized score by over-emphasizing only part of the objective or by moving away from the scaffold-local neighborhood, resulting in unstable optimization and rapidly degraded success rates as task complexity increases.

The comparison between SFT and DPO further reveals that preference alignment strengthens this extrapolative generalization ability. Across all four unseen three-property tasks, DPO consistently improves SR and RI over SFT, often by a large margin, while moderately reducing similarity. In other words, SFT variants are more conservative and stay closer to the source scaffold, whereas DPO pushes the model toward stronger multi-property optimization by better combining attribute-level optimization signals. This pattern suggests that DPO does not simply overfit to the seen objectives; rather, it enhances the model's ability to recombine lower-order supervision into effective higher-order edits.

Overall, these results support two conclusions. First, lower-order supervision already contains reusable optimization primitives that can be composed to solve unseen higher-order objectives. Second, our alignment

Table 8: Compact summary of SCPT failure patterns.

| Setting | Avg. SR | Dominant failure pattern | Interpretation |
|---|---|---|---|
| Single-property tasks | 93.3% | Invalid generations | The model usually learns the target direction, but some local edits produce invalid or non-evaluable molecules. |
| Multi-property tasks | 69.4% | Wrong direction / insufficient improvement | The main difficulty becomes satisfying multiple directional property constraints simultaneously. |

framework, especially DPO, improves the model's ability to perform this compositional extrapolation while maintaining a much more scaffold-preserving optimization regime than traditional non-LLM baselines. These results should be interpreted as evidence of compositional extrapolation under controlled oracle-defined objectives, rather than as a complete solution to real-world lead optimization. Practical campaigns often involve additional and potentially conflicting constraints, such as potency, solubility, selectivity, toxicity, synthetic accessibility, and ADMET endpoints. Extending SCPT to such settings may require Pareto-aware preference construction, constraint-conditioned prompts, controllable objective weights, and active data construction for rare high-order objective combinations.

### 4.7 Failure Modes and Practical Scope

To better characterize when SCPT fails, we categorize failed generations into three types: Invalid, Wrong Direction, and Insufficient Improvement. Invalid means that no generated candidate can be parsed or evaluated by the oracle. Wrong Direction means that at least one target property changes in the opposite direction. Insufficient Improvement means that the direction is correct but the gain does not reach the task-specific threshold.

This analysis indicates that SCPT is comparatively reliable for single-property local optimization, where most failures are molecular validity issues. In multi-property optimization, the dominant failure mode shifts from validity to directional control, suggesting that the model struggles mainly when several property directions and thresholds must be satisfied at the same time. Detailed task-level and model-level analyses are provided in Appendix D.

## 5  Conclusion

We propose a **scaffold-conditioned preference triplet** pipeline for molecular optimization that treats molecular strings as a structured language and aligns LLM-based editors with chemistry-grounded preferences. Our pipeline systematizes dispersed medicinal-chemistry heuristics into a reproducible data curation process: scaffold-conditioned alignment and chemistry-driven filters for validity, synthesizability, and meaningful property gains produce triplets ⟨scaffold, better, worse⟩ that encode relative preference within a scaffold neighborhood. Using these preferences for post-training, models achieve property-improving edits while better preserving scaffold and local structure on application-oriented benchmarks. Beyond in-family comparisons, we further show that LLMs trained under this paradigm outperform representative non-LLM molecular optimization baselines in scaffold-constrained and multi-objective settings, indicating a stronger ability to perform structurally faithful local optimization. Moreover, the learned optimization behavior generalizes beyond directly observed supervision: models trained on single-property and two-property data remain effective on unseen three-property optimization tasks, suggesting promising extrapolative generalization to higher-order objective combinations. Ablation studies show that similarity thresholds, property-gap filters, and DPO hyperparameters jointly induce a predictable frontier among success rate, improvement, and scaffold similarity, enabling systematic control over the gain–locality trade-off. Overall, unifying chemical priors, data curation, and alignment yields a stronger and more controllable paradigm for preference-based molecular optimization, with particular advantages in scaffold preservation, multi-objective optimization, and compositional generalization.

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

## A   Algorithm

The SCPT pipeline and the alignment framework are illustrated in Algorithm 1.

## B   Data Construction

For all properties considered in this work, we construct scaffold-conditioned preference triples from molecules sampled from the ZINC/ZINC250k library Sterling & Irwin (2015); Gómez-Bombarelli et al. (2018). For each property, we first annotate candidate molecules using oracles Gao et al. (2022b); Huang et al. (2021); Li et al. (2024) and form ordered pairs $(x, y)$ such that $y$ is better than $x$ with respect to the desired optimization direction (higher or lower). We apply a preliminary numeric filter by retaining only pairs whose property difference lies in the upper half of the distribution, which reduces the cost of subsequent structural checks. We then keep pairs with Tanimoto similarity $\geq 0.6$ and use a graph-based edit check to ensure exactly one local modification between $x$ and $y$ (a single bond break or substitution). We consider eight standard single-property tasks:

**pLogP**: penalized logP capturing lipophilicity and related ADMET factors; **QED**: quantitative estimate of drug-likeness; **BBBP**: blood–brain barrier permeability; **Mutag**: the ability of a compound to induce genetic mutations; **HIA**: human intestinal absorption; **DRD2**: binding affinity to the dopamine D2 receptor; **JNK3**: c-Jun N-terminal kinase 3 binding affinity; **GSK3$\beta$**: glycogen synthase kinase $3\beta$ inhibitory activity. Following prior work Dey et al. (2025), we impose property-specific difference thresholds to ensure practically meaningful improvements: an increase $> 1.0$ for penalized LogP (pLogP), increases $> 0.1$ for QED, increase $> 0.05$ HIA, an increase $> 0.2$ for BBBP and DRD2, and a decrease $> 0.1$ for Mutag (i.e., $\Delta\text{Mutag} \leq -0.1$). For the kinase-focused tasks JNK3 (c-Jun N-terminal kinase 3) and GSK3$\beta$ (glycogen synthase kinase-3 beta), which quantify inhibitory activity toward clinically relevant targets, we use an improvement threshold of 0.05 for both properties. These filtered pairs are then converted into scaffold-conditioned $\langle\text{scaffold}, \text{better}, \text{worse}\rangle$ triplets using the pipeline described in Section 3.

## C   Experiment

### C.1   Metrics

**(1) Success Rate (SR).** The fraction of source molecules for which the optimization succeeds:

$$\text{SR} = \frac{N_{\text{success}}}{N_{\text{total}}} \tag{12}$$

**Algorithm 1** SCPT Pipeline and Alignment Framework

**Input:** Molecular database $\mathcal{M}$, Large Language Model $\mathcal{L}$, ADMET predictor $\mathcal{A}$, property threshold $\delta_{\text{prop}}$, similarity threshold $\gamma$

**Output:** Optimized LLM with LoRA parameters $\mathcal{L}^*$

1: Initialize empty sets: $\mathcal{SCPT}_{\text{thre}} \leftarrow \emptyset$, $\mathcal{SCPT}_{\text{SFT}} \leftarrow \emptyset$, $\mathcal{SCPT}_{\text{DPO}} \leftarrow \emptyset$
2: $\mathcal{SCPT}_{\text{pair}} \leftarrow$ pairs in $\mathcal{M}$
3: **for** each pair $p \in \mathcal{SCPT}_{\text{pair}}$ **do**
4:     $(m_{sou}, m_{opt}) \leftarrow p$
5:     $\text{prop}_{sou} \leftarrow \mathcal{A}(m_{sou})$, $\text{prop}_{opt} \leftarrow \mathcal{A}(m_{opt})$
6:     $\text{sim} \leftarrow \text{TANIMOTO}(m_{sou}, m_{opt})$
7:     **if** $|\text{prop}_{opt} - \text{prop}_{sou}| > \delta_{\text{prop}}$ **and** $\text{sim} > \gamma$ **then**
8:         $\mathcal{SCPT}_{\text{thre}} \leftarrow \mathcal{SCPT}_{\text{thre}} \cup \{p\}$
9:     **end if**
10: **end for**
11: **for** each pair $p \in \mathcal{SCPT}_{\text{thre}}$ **do**
12:     **if** $\text{DFGED}(p) = 1$ **then**
13:         $\mathcal{SCPT}_{\text{SFT}} \leftarrow \mathcal{SCPT}_{\text{SFT}} \cup \{p\}$
14:     **end if**
15: **end for**
16: **for** each pair $p \in \mathcal{SCPT}_{\text{SFT}}$ **do**
17:     $\text{MSC} \leftarrow \text{RDKIT.GETMCS}(p)$
18:     $\mathcal{T} \leftarrow \langle \text{MSC}, p \rangle$
19:     $\mathcal{SCPT}_{\text{DPO}} \leftarrow \mathcal{SCPT}_{\text{DPO}} \cup \{\mathcal{T}\}$
20: **end for**
21: $\mathcal{L}^* \leftarrow \text{OPTIMIZELORA-SFT}(\mathcal{L}, \mathcal{SCPT}_{\text{SFT}})$
22: $\mathcal{L}^* \leftarrow \text{OPTIMIZELORA-DPO}(\mathcal{L}^*, \mathcal{SCPT}_{\text{DPO}})$
23: **return** $\mathcal{L}^*$

Table 9: Summary statistics of single-property training data.

|        | Mutagenicity | pLogP  | QED    | BBBP  | HIA   | GSK3$\beta$ | JNK3  | DRD2   |
|--------|--------------|--------|--------|-------|-------|-------------|-------|--------|
| N      | 110586       | 105949 | 157863 | 42296 | 13929 | 47168       | 22711 | 116755 |
| mean   | -0.21        | 1.88   | 0.18   | 0.32  | 0.34  | 0.17        | 0.10  | 0.40   |
| std    | 0.10         | 1.52   | 0.09   | 0.11  | 0.21  | 0.07        | 0.08  | 0.17   |
| var    | 0.01         | 2.32   | 0.01   | 0.01  | 0.05  | 0.01        | 0.01  | 0.03   |
| min    | -0.85        | 1.00   | 0.10   | 0.20  | 0.10  | 0.11        | 0.06  | 0.20   |
| q1     | -0.26        | 1.28   | 0.12   | 0.24  | 0.16  | 0.12        | 0.06  | 0.26   |
| median | -0.18        | 1.59   | 0.15   | 0.29  | 0.27  | 0.14        | 0.08  | 0.36   |
| q3     | -0.13        | 1.91   | 0.21   | 0.37  | 0.45  | 0.19        | 0.11  | 0.51   |
| max    | -0.10        | 54.95  | 0.77   | 0.92  | 0.99  | 0.89        | 0.77  | 1.00   |

**(2) Tanimoto Similarity (SIM)** The structural similarity between the optimized molecule and its source counterpart, reflecting scaffold preservation. It is computed as in Eq. (6) of the paper.

**(3) Relative Improvement (RI).** The average relative change across multiple target properties. Let $\mathcal{P}$ be the set of properties. Then

$$\text{RI} = \frac{1}{|\mathcal{P}|} \sum_{p \in \mathcal{P}} \text{RI}_p \tag{13}$$

where $RI_p$ is computed as:

$$\text{RI}_p = \frac{\mathbb{D}[p]\big(p(m_{opt}) - p(m_{sou})\big)}{p(m_{sou})} \tag{14}$$

Table 10: Summary statistics of multi-property training data.

| Properties | Count | Properties | Count |
|---|---|---|---|
| GSK3$\beta$+DRD2+pLogP | 9,469 | JNK3+QED | 8,152 |
| GSK3$\beta$+DRD2+QED | 13,898 | JNK3+pLogP | 10,609 |
| GSK3$\beta$+pLogP+QED | 7,268 | GSK3$\beta$+QED | 23,251 |
| DRD2+pLogP+QED | 2,349 | GSK3$\beta$+pLogP | 19,571 |
| JNK3+GSK3$\beta$ | 9,885 | pLogP+QED | 15,874 |

where $\mathbb{D}[p] \in \{+1, -1\}$ encodes the desired direction for property $p$ ($+1$ if larger is better, $-1$ if smaller is better), and $p(m_{sou})$ and $p(m_{opt})$ denote the values of property $p$ for the source molecule $m_{sou}$ and the generated molecule $m_{opt}$, respectively. Unless otherwise specified, all three metrics are "higher is better".

## C.2 Models and Baselines

We briefly describe the baseline language models used in our experiments.

**Closed-source general LLM  ChatGPT-4o** Achiam et al. (2023) is OpenAI's flagship GPT-4 series model and a proprietary, large-scale multimodal LLM. It is trained on a mixture of public and proprietary data and optimized for a wide range of tasks, including reasoning, dialogue, coding, and instruction following. In our experiments, we access ChatGPT-4o via the official text-completion API and restrict ourselves to the text-only interface (i.e., we do not use any vision capabilities). We treat ChatGPT-4o as a strong general-purpose baseline that is not specifically tuned for chemistry.

**Claude-3.5** Anthropic (2025) is Anthropic's frontier Claude-3 family model, designed as a high-capability general-purpose assistant for reasoning, coding, and complex workflows. Similar to ChatGPT-4o, Claude-3.5 is proprietary and trained on large-scale mixed data, with safety and alignment techniques such as constitutional AI. We use the text-completion API and evaluate Claude-3.5 under the same prompting protocols as ChatGPT-4o, without any chemistry-specific fine-tuning.

**Open-source general LLMs.  LLaMA-3.1-8B** Dubey et al. (2024) is an 8B-parameter open-weight model released by Meta. It is pre-trained on a large, diverse corpus (including web pages, code, and other text) and further instruction-tuned to follow natural language commands and engage in dialogue. We use the instruction-tuned variant as a representative open-source general-purpose model with a relatively modest parameter count. Unless otherwise noted, we employ the default chat/instruction interface and decode with temperature and sampling settings matched to other models.

**Mistral-2.0-Instruct-7B** Jiang et al. (2023) is a 7B-parameter instruction-tuned model from Mistral AI. It builds on the Mistral architecture with efficient attention and strong performance despite its small size. The Instruct variant is optimized for following user prompts and is widely used as a compact open-source baseline for downstream tasks. In our experiments, we treat Mistral-2.0-Instruct-7B as another competitive general-purpose LLM that has not received any chemistry-specific adaptation.

**Chemistry-specialized LLMs.  ChemLLM** Zhang et al. (2024) is a chemistry-focused language model that extends a general-purpose backbone with large-scale pretraining and instruction tuning on chemistry-related corpora. The training data typically includes molecular and reaction text, SMILES strings, patents, and curated chemistry question–answer pairs. As a result, ChemLLM is better calibrated for chemical nomenclature, molecular structure manipulation, and property-related reasoning than purely general-purpose LLMs. We use the instruction-tuned ChemLLM variant and query it in a zero-shot fashion on our molecular optimization tasks.

$LLaSMol_{Mistral}$ Yu et al. (2024) is a member of the LLaSMol family of SMILES-centric chemistry models built on top of the Mistral architecture. It is instruction-tuned on a large collection of molecule-level tasks, such as property prediction, forward and retrosynthesis, and molecular editing, where both natural language instructions and SMILES strings are used as supervision. Compared to generic Mistral-2.0-Instruct-7B,

$LLaSMol_{Mistral}$ incorporates domain-specific priors and task formats tailored to chemistry. We use the released checkpoint and follow the recommended inference configuration, evaluating it in the zero-shot setting on the same prompts as other models.

For all LLM baselines, we adopt a unified prompting protocol for property optimization, with task descriptions and input molecules expressed in natural language and SMILES. General-purpose LLMs (ChatGPT-4o, Claude-3.5, LLaMA-3.1-8B, and Mistral-2.0-Instruct-7B) are evaluated under both zero-shot and five-shot settings, whereas chemistry-specialized models (ChemLLM and $LLaSMol_{Mistral}$) are evaluated in the zero-shot setting for a fair comparison to their typical usage scenario.

**Non-LLM baseline**  As representative non-LLM baselines, we consider GRAPHGA, JTVAE, REIN-VENT and GENMOL, which cover three widely used paradigms in molecular optimization. GRAPHGA is a graph-based genetic algorithm that searches molecular space through mutation and crossover operations, making it a representative evolutionary optimization method. JTVAE is a junction-tree variational autoencoder that encodes molecules into a structured latent space and performs optimization by searching within that latent space, representing latent-variable generative approaches for molecular design. REINVENT is a reinforcement-learning-based SMILES generator that updates a pretrained policy toward molecules with higher task rewards, and is widely used as a strong policy-optimization baseline in molecular generation. GENMOL is a discrete-diffusion molecular generation model based on SMILES molecular representations. Together, these baselines provide complementary non-LLM comparisons spanning evolutionary search, latent-space optimization, and reinforcement learning.

### C.3  Experiment Setup

### C.3.1  Experimental Setup for LLMs

We adopt two backbone LLMs with strong community performance and adoption in chemistry applications: **Mistral-2.0-Instruct-7B** and **LLaMA-3.1-8B**. We fine-tune these models with LoRA Hu et al. (2022). During the SFT stage, we use a next-token objective with prompts that concatenate the source molecule and property-optimization instructions, training the model to output the SMILES of an optimized molecule. We then perform DPO post-training on our ⟨scaffold, better, worse⟩ triplet dataset to further align preferences for scaffold-conditioned, directionally guided edits.

For both stages, we apply parameter-efficient fine-tuning with LoRA adapters of rank $r = 8$, scaling factor $\alpha = 16$, and LoRA dropout 0.05, attached to the attention projections and MLP layers . We train with an effective batch size of 128, implemented via per-device micro-batch sizes of 16 for SFT and 8 for DPO with gradient accumulation, for 3 epochs in both SFT and DPO. Optimization uses AdamW with a cosine learning-rate schedule and a warmup ratio of 0.1. The SFT stage uses a learning rate of $1 \times 10^{-4}$, while the DPO stage uses a smaller learning rate of $5 \times 10^{-6}$ and a DPO temperature parameter of $\beta = 0.5$. We train in bfloat16 precision. Following standard practice for DPO, we instantiate a frozen reference model with the same backbone and LoRA configuration and optimize only the policy model parameters.

For single-property tasks, we construct task-specific training and test sets: for the experiments in RQ1, we subsample 2,000 molecule pairs for each task to enable controlled analysis. For RQ2 and RQ3, we use the full set of valid training samples that satisfy the corresponding construction criteria. To prevent data leakage, none of the test molecules appear in the corresponding training set. For multi-property tasks, due to limited data, we merge data from all tasks and train a single model once. We then evaluate each task on its own test set of 500 molecules, with all test molecules excluded from the training set.

At inference time, for each source molecule the model generates 20 candidate molecules. We score candidates with the property oracle and select the best-performing one as the final optimized result for metric computation. For a source molecule $m$ with candidates $\{m'_k\}_{k=1}^{20}$, the selected output is

$$\hat{m} = \arg\max_{k \in \{1, \ldots, 20\}} F(m'_k) \tag{15}$$

where $F(\cdot)$ denotes the task-specific property objective (or a weighted multi-property score, when applicable).

Table 11: Training cost and infrastructure for SCPT alignment.

| Stage | Backbone | Hardware | Epochs | Batch / Micro-batch | Wall-clock time |
|-------|----------|----------|--------|---------------------|-----------------|
| SFT | LLaMA-3.1-8B / Mistral-2.0-Instruct-7B | 1× RTX 4090 24GB | 3 | 128 / 16 | ∼45 min |
| DPO | LLaMA-3.1-8B / Mistral-2.0-Instruct-7B | 1× RTX 4090 24GB | 3 | 128 / 8 | ∼95 min |

### C.3.2 Training cost and infrastructure

Unless otherwise specified, the remaining optimization and decoding hyperparameters follow the settings described in Section 4.1 and Appendix C.

### C.3.3 Experimental Setup for Compositional Extrapolation

To study compositional extrapolation under limited high-order supervision, we construct the training data for this experiment using only *single-property* and *pairwise-property* optimization instances derived from four properties: QED, pLogP, GSK3$\beta$, and DRD2. Concretely, we collect training examples from threshold-qualified data. We then retain only the single-property and pairwise-property subsets for training, while reserving all three-property combinations exclusively for evaluation.

This design is motivated by the data sparsity observed during SCPT construction. As the number of jointly optimized properties increases, it becomes increasingly difficult to obtain valid training pairs that simultaneously satisfy all directional property requirements while also remaining within the scaffold-preserving local neighborhood induced by our similarity and structural filters. As a result, the amount of directly usable supervision drops rapidly for higher-order objectives. The extrapolation setting therefore tests whether the model can acquire reusable optimization primitives from lower-order supervision and recombine them at test time to solve unseen higher-order tasks.

The four unseen three-property evaluation tasks are: **DPQ**: DRD2 + pLogP + QED; **GDP**: GSK3$\beta$ + DRD2 + pLogP; **GBD**: GSK3$\beta$ + QED + DRD2; **GQP**: GSK3$\beta$ + QED + pLogP. Unless otherwise specified, all other training settings follow the main experiments, including the same model backbones, prompting format, SCPT construction pipeline, SFT configuration, DPO hyperparameters, decoding strategy, and evaluation protocol.

### C.3.4 Experimental Setup for Non-LLM Baselines

For both single-property and multi-property comparisons, we evaluate representative non-LLM baselines under the PMO framework, including GraphGA, JTVAE, REINVENT and GenMol. To make these methods better aligned with the scaffold-preserving objective studied in this work, we modify their optimization targets by explicitly incorporating similarity to the source molecule.

For single-property tasks, the optimization target is defined as

$$y = \text{SIM}(x, x_0) + s(x),$$

where $x_0$ is the source molecule, $x$ is the generated molecule, $\text{SIM}(x, x_0)$ denotes the structural similarity between $x$ and $x_0$, and $s(x)$ is the task-specific property score. For multi-property tasks, we extend the target to

$$y = \text{SIM}(x, x_0) + \sum_{p \in \mathcal{T}} s_p(x),$$

where $\mathcal{T}$ is the set of target properties and $s_p(x)$ is the oracle score for property $p$. In practice, this modification is applied to the fitness function in GraphGA, the surrogate-model optimization target in JTVAE, and the reward function in REINVENT. Therefore, all three baselines are encouraged to jointly optimize structural similarity and the target property (or the sum of target properties in the multi-objective setting).

Unless otherwise specified, all remaining settings follow the original PMO framework, including the starting molecule pool, oracle definitions, pretrained initialization, and the overall evaluation protocol. This ensures

that the comparison focuses on the effect of introducing similarity-aware objectives, rather than changing the underlying training or search pipeline.

For GRAPHGA, we use the PMO implementation with Bayesian hyperparameter search. We tune the main search-related parameters, including the population size, offspring size, and mutation rate. In particular, the population size is searched in the range $[50, 200]$, the offspring size in $[50, 300]$, and the mutation rate in $[0, 0.1]$. These parameters mainly control the exploration strength and diversity of the genetic search process.

For JTVAE, we adopt the fast PMO configuration initialized from the pretrained checkpoint provided in the framework. We keep the standard model configuration, with hidden size 450, latent size 56, tree depth 20, and graph depth 3. Following PMO, optimization is performed in the latent space with a lightweight Bayesian optimization procedure, where the number of optimization rounds is set to 20 and the batch size of each BO step is 10.

For REINVENT, we also use the PMO implementation and initialize the policy from the default pretrained prior. The main hyperparameters are selected by grid search over a small set of key reward-learning parameters, including batch size, sigma, and experience replay size. Specifically, the batch size is searched over $\{32, 64\}$, sigma over $\{250, 500, 750, 1000\}$, and the experience replay size over $\{8, 16, 24, 32\}$. These parameters mainly determine the strength and stability of policy optimization under the modified reward.

For GENMOL, we adopt the official implementation with the released pretrained checkpoint and default generation configuration provided by the authors.

Overall, the purpose of this setup is not to redesign these baselines, but to adapt them as fairly as possible to the source-conditioned, scaffold-preserving optimization setting. By injecting similarity into their optimization targets while otherwise keeping the PMO framework intact, we obtain non-LLM comparisons that are more appropriate for evaluating whether our LLM-based approach offers a genuine advantage under structural preservation constraints.

### C.4 Prompt Templates

In this section, we document the exact prompt templates used in the SFT and DPO stages. All prompts instruct the model to behave as an expert medicinal chemist, to make minimal modifications to the input molecule while preserving its overall structure, and to return only a SMILES string wrapped in special `<smiles>` delimiters.

**SFT: single-property optimization.** For single-property optimization, we use the following instruction-style template, where `{source_smiles}` is the input molecule and `{property}` is the property to be increased or decreased:

> You are an expert medicinal chemist.
> Given a source molecule and a desired property change, modify the molecule as little as possible while preserving its overall structure, and generate a new molecule that satisfies the requirement. Return only the molecule as a SMILES string wrapped in `<smiles>` tags, with no additional text.
>
> Request: Given the source molecule `<smiles>{source_smiles}<smiles>`, {increase/decrease} {property}.
> Answer: `<smiles>{target_smiles}<smiles>`

**SFT: multi-property optimization.** For multi-property optimization, we extend the same template to list multiple property requirements in the request:

> You are an expert medicinal chemist.
> Given a source molecule and several desired property changes, modify the molecule as little as possible while preserving its overall structure, and generate a new molecule that satisfies all of the requirements. Return only the molecule as a SMILES string wrapped in `<smiles>` tags, with no additional text.
>
> Request: Given the source molecule `<smiles>{source_smiles}<smiles>`, {increase/decrease}

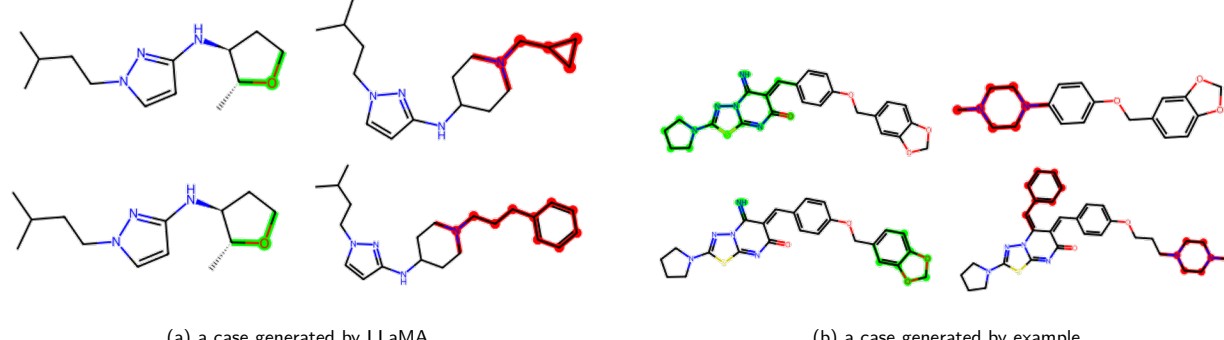

(a) a case generated by LLaMA

(b) a case generated by example

Figure 5: Molecular pair examples optimized for the DRD2 property. Green denotes the original molecule, while red denotes the optimized molecule. (a) shows the SFT-optimized results, (b) shows the DPO-optimized results.

> {property_1}, {increase/decrease} {property_2}, {increase/decrease} {property_3}, ...
> Answer: `<smiles>{target_smiles}<smiles>`

**DPO: scaffold-conditioned preference triples.** For DPO, each ⟨scaffold, better, worse⟩ triple is converted into a shared prompt and two alternative completions (*chosen* and *rejected*). The prompt uses the scaffold as the source molecule and lists the desired property changes:

> You are an expert medicinal chemist.
> Given a source molecule and several desired property changes, modify the molecule as little as possible while preserving its overall structure (scaffold), and generate a new molecule that satisfies all of the requirements. Return only the molecule as a SMILES string wrapped in `<smiles>` tags, with no additional text.
>
> Request: Given the source molecule `<smiles>{scaffold_smiles}<smiles>`, {increase/decrease} {property_1}, {increase/decrease} {property_2}, {increase/decrease} {property_3}, ...

The corresponding *chosen* and *rejected* responses are instantiated as:

> Accepted (chosen): `<smiles>{better_smiles}<smiles>`
> Rejected (rejected): `<smiles>{worse_smiles}<smiles>`

In all cases, the model is instructed to output only the `<smiles>`–delimited SMILES string, without any explanation or additional text.

## C.5 Case study

As shown in Figure 5, the single-property case study indicates that, compared to the SFT-trained LLM, the DPO-aligned LLM tends to make larger structural edits and explore a broader region of chemical space, leading to better property improvements while still preserving molecular similarity.

As shown in Figure 6, this panel visualizes parameter sensitivity on the BPQ task.. Panels (a–d) use decreasing learning rates; within each panel, the three rows correspond to $\beta$=0.1 (top), 0.3 (middle), and 0.5 (bottom). As the learning rate decreases, the extent of edits shrinks and property gains diminish; at a fixed learning rate, larger $\beta$ further suppresses the edit magnitude and slightly increases similarity.

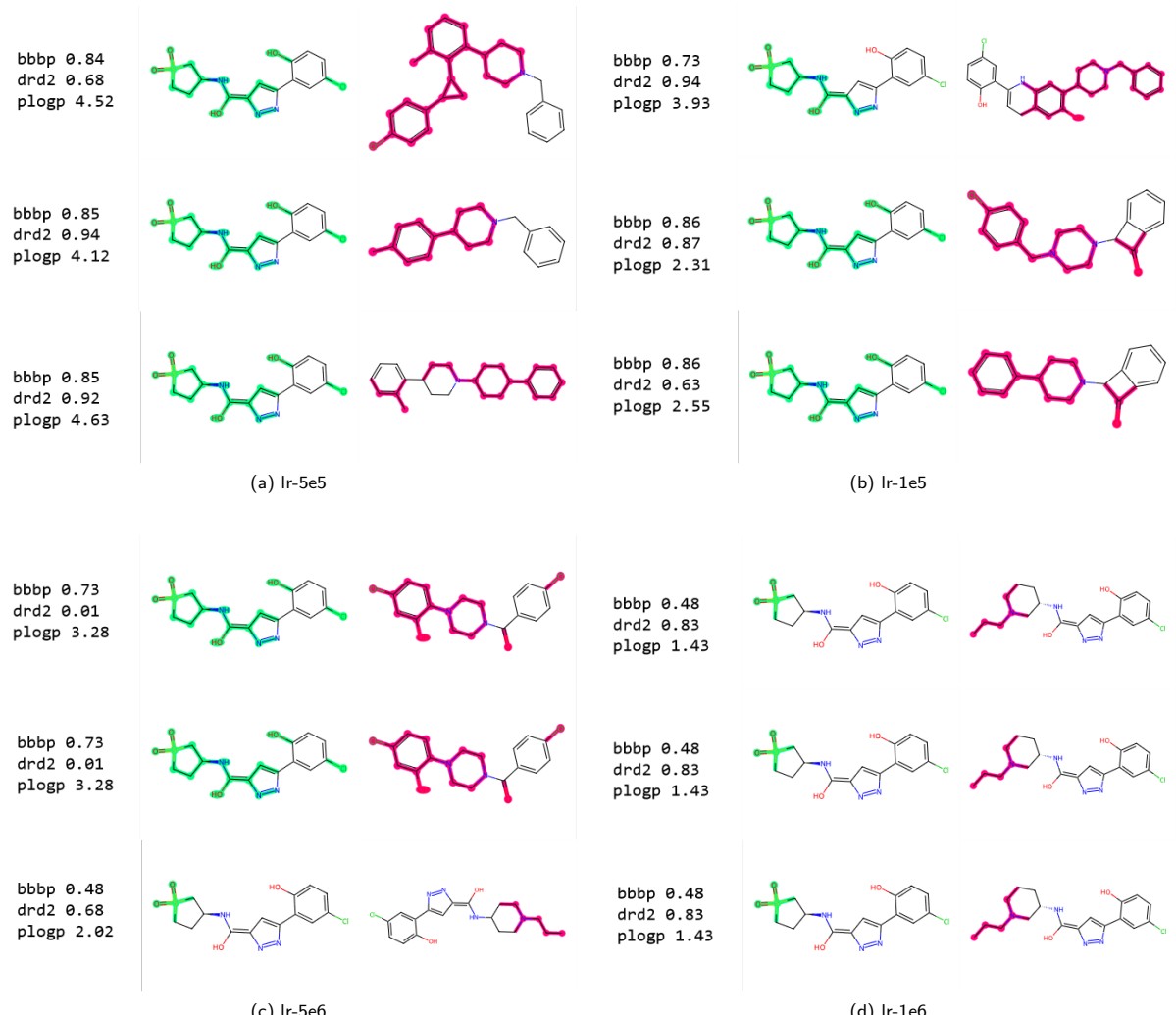

Figure 6: Molecular pair example optimized with different hyperparameter configurations. In the subfigure, the three rows (top to bottom) correspond to the original and optimized molecules at $\beta = 0.1$, 0.3 and 0.5 respectively. Green denotes the original molecule, while red denotes the optimized molecule.

### C.6 Detailed Analysis And Result

### C.6.1 Similarity Control

To examine how the similarity threshold affects training, we bin the Tanimoto similarity into six ranges (0.3–0.4, 0.4–0.5, 0.5–0.6, 0.6–0.7, 0.7–0.8, 0.8–0.9) and compare eight single-property tasks (BBBP, DRD2, HIA, Mutag, pLogP, QED, GSK3$\beta$, JNK3). For each property-by-bin condition, we sample 2,000 molecular pairs and match the distribution of property differences across bins using the high-similarity bin as a template, so that the similarity threshold is the only varying factor, Density distributions of property differences in the training data are shown in Figure 7.

The trends are consistent across tasks. Most endpoints maintain high SR with limited sensitivity to similarity, whereas DRD2 drops as the threshold tightens, indicating a stronger dependence on the available edit space. SIM increases with the threshold but not linearly; it plateaus around 0.5–0.6, where scaffold fidelity is already well preserved and further tightening yields marginal gains. In contrast, RI decreases as the threshold increases: stronger similarity constraints compress the learnable local edits, shifting from multi-site changes

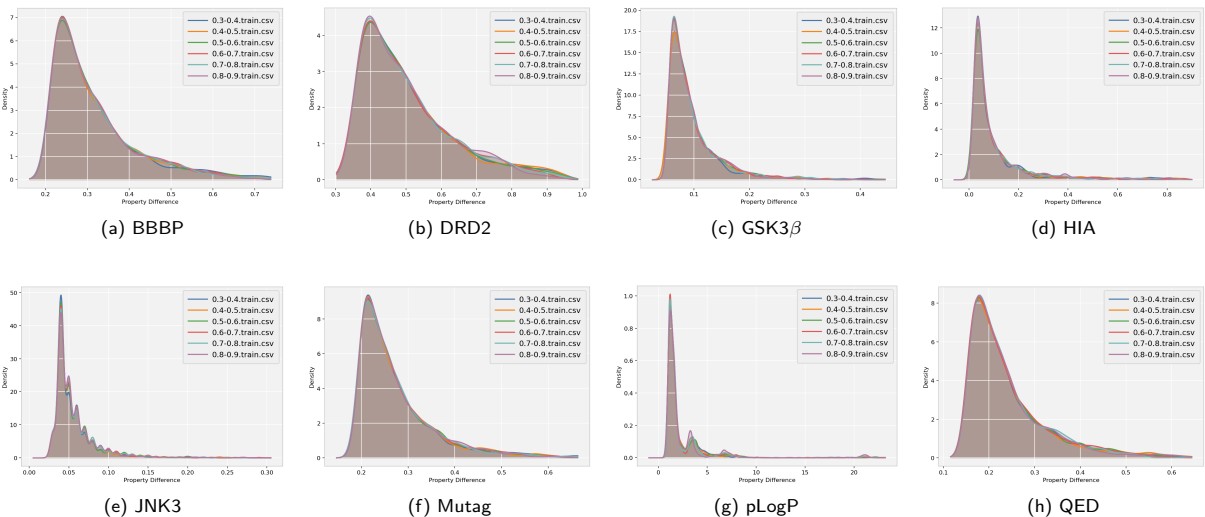

Figure 7: Density of Property Differences in Training Data Across Tasks Under Different Similarity Conditions. The figure presents the kernel density distributions of property differences in the training dataset for each task.

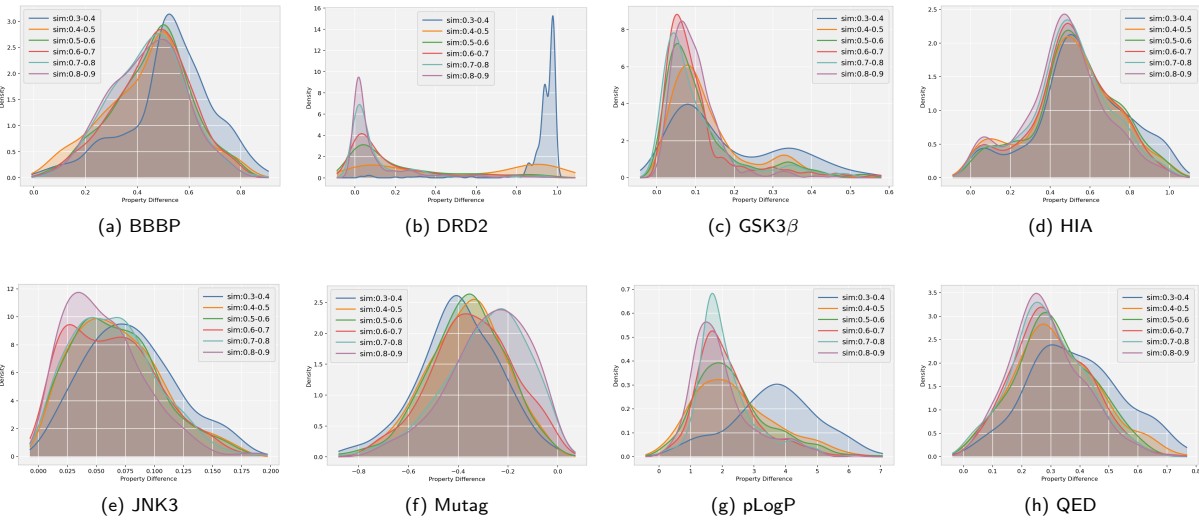

Figure 8: Density of Property Differences Between Source and Optimized Molecules Across Tasks under different similarity condition. The figure shows kernel density distributions of the property difference for each task. Subfig (f) is a reduction-oriented task, so its distribution trends in the opposite direction (toward negative values).

toward near single-site modifications and thereby reducing achievable improvements. This pattern holds broadly, with Mutag showing milder variation. The detailed property density distributions are shown in Figure 8 and the complete experimental results are reported in Table 12.

Overall, the similarity threshold induces a stable trade-off between similarity and improvability. We recommend operating around 0.5–0.6 to balance scaffold preservation and property gains; stricter thresholds can be used when fidelity is paramount, with the caveat of reduced RI and potential SR drops on sensitive endpoints such as DRD2.

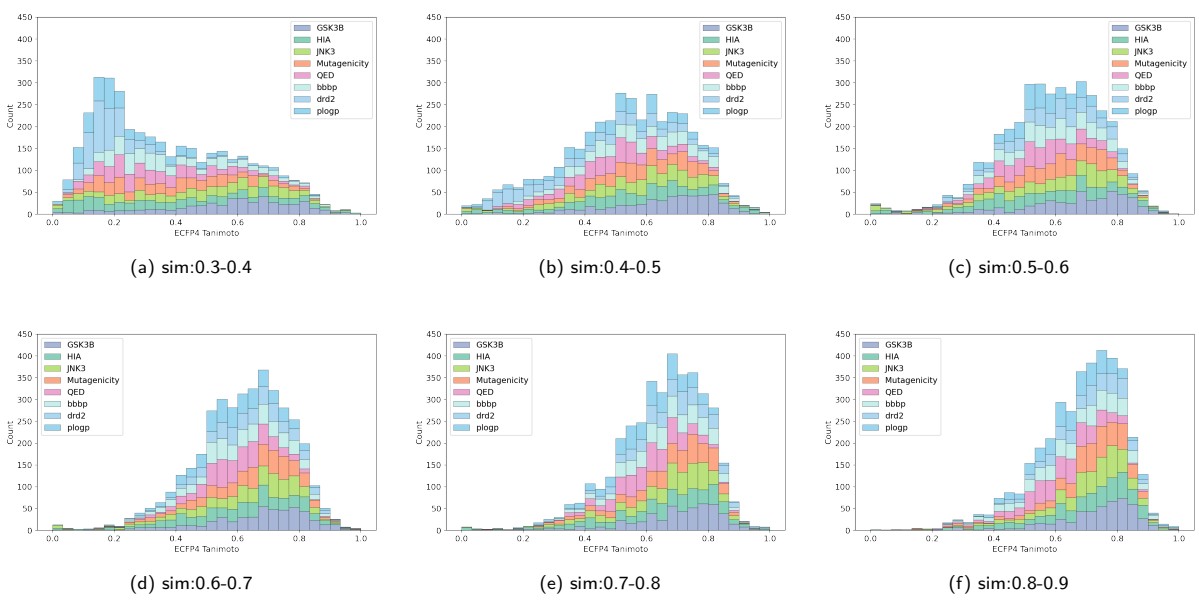

Figure 9: Distribution of Tanimoto similarity between optimized molecules and original molecules under different similarity conditions.

Table 12: Single-property optimization results by similarity bin. Rows indicate similarity intervals and columns correspond to tasks.

| Similarity | JNK3 | | | BBBP | | | DRD2 | | | HIA | | | Mutag↓ | | | pLogP | | | QED | | | GSK3β | | |
|---|---|---|---|---|---|---|---|---|---|---|---|---|---|---|---|---|---|---|---|---|---|---|---|---|
| | SR | SIM | RI | SR | SIM | RI | SR | SIM | RI | SR | SIM | RI | SR | SIM | RI | SR | SIM | RI | SR | SIM | RI | SR | SIM | RI |
| 0.3–0.4 | 93.0 | 0.48 | 2.94 | 98.8 | 0.39 | 1.83 | 99.8 | 0.19 | 25.55 | 98.2 | 0.41 | 5.09 | 97.6 | 0.39 | 0.59 | 97.6 | 0.27 | 12.98 | 99.0 | 0.35 | 1.10 | 93.4 | 0.57 | 7.25 |
| 0.4–0.5 | 95.8 | 0.57 | 2.39 | 98.4 | 0.57 | 1.50 | 96.0 | 0.41 | 12.87 | 97.6 | 0.56 | 3.76 | 98.0 | 0.57 | 0.55 | 96.4 | 0.50 | 8.99 | 99.4 | 0.49 | 0.88 | 97.4 | 0.65 | 5.32 |
| 0.5–0.6 | 96.6 | 0.59 | 2.31 | 99.8 | 0.59 | 1.56 | 89.0 | 0.60 | 6.08 | 97.6 | 0.57 | 3.83 | 98.2 | 0.60 | 0.55 | 95.4 | 0.56 | 9.26 | 99.8 | 0.55 | 0.87 | 97.4 | 0.66 | 4.60 |
| 0.6–0.7 | 97.2 | 0.62 | 2.23 | 99.6 | 0.61 | 1.57 | 84.4 | 0.64 | 5.28 | 97.6 | 0.61 | 3.60 | 99.6 | 0.64 | 0.52 | 98.8 | 0.60 | 8.44 | 99.8 | 0.60 | 0.83 | 99.2 | 0.67 | 4.05 |
| 0.7–0.8 | 99.0 | 0.66 | 2.27 | 100.0 | 0.65 | 1.49 | 82.8 | 0.68 | 4.12 | 99.4 | 0.65 | 3.41 | 99.4 | 0.67 | 0.44 | 99.6 | 0.61 | 8.46 | 99.6 | 0.61 | 0.80 | 99.2 | 0.70 | 4.08 |
| 0.8–0.9 | 98.6 | 0.70 | 1.90 | 100.0 | 0.66 | 1.47 | 76.0 | 0.71 | 3.34 | 99.6 | 0.71 | 3.31 | 98.6 | 0.71 | 0.42 | 99.2 | 0.66 | 7.72 | 99.4 | 0.62 | 0.79 | 99.8 | 0.73 | 3.62 |

Table 13: Single-property optimization results by property-gap thresholds. Each row corresponds to a property-difference percentile range (the first row is the top 10% largest gaps), and each column to a task.

| Property Range | BBBP | | | DRD2 | | | HIA | | | Mutag | | | pLogP | | | QED | | | GSK3β | | | JNK3 | | |
|---|---|---|---|---|---|---|---|---|---|---|---|---|---|---|---|---|---|---|---|---|---|---|---|---|
| | SR↑ | SIM↑ | RI↑ | SR↑ | SIM↑ | RI↑ | SR↑ | SIM↑ | RI↑ | SR↑ | SIM↑ | RI↑ | SR↑ | SIM↑ | RI↑ | SR↑ | SIM↑ | RI↑ | SR↑ | SIM↑ | RI↑ | SR↑ | SIM↑ | RI↑ |
| 0–0.1 | 100.0 | 0.66 | 1.48 | 76.0 | 0.67 | 3.88 | 99.4 | 0.64 | 3.75 | 98.8 | 0.68 | 0.45 | 99.8 | 0.60 | 8.97 | 99.6 | 0.61 | 0.79 | 99.6 | 0.70 | 5.17 | 99.6 | 0.65 | 2.76 |
| 0.1–0.2 | 100.0 | 0.66 | 1.49 | 76.4 | 0.68 | 4.12 | 99.4 | 0.66 | 3.37 | 99.6 | 0.66 | 0.49 | 99.4 | 0.63 | 7.69 | 99.6 | 0.62 | 0.77 | 98.0 | 0.72 | 2.71 | 98.2 | 0.68 | 1.49 |
| 0.2–0.3 | 100.0 | 0.65 | 1.45 | 79.6 | 0.69 | 3.30 | 99.0 | 0.66 | 3.55 | 99.8 | 0.67 | 0.48 | 99.8 | 0.64 | 6.57 | 99.8 | 0.63 | 0.70 | 98.6 | 0.73 | 2.74 | 96.4 | 0.67 | 0.98 |
| 0.3–0.4 | 100.0 | 0.67 | 1.37 | 75.8 | 0.70 | 2.92 | 99.4 | 0.66 | 3.36 | 99.6 | 0.67 | 0.48 | 99.8 | 0.64 | 6.46 | 99.4 | 0.65 | 0.66 | 97.4 | 0.72 | 2.20 | 95.2 | 0.70 | 0.96 |
| 0.4–0.5 | 100.0 | 0.66 | 1.36 | 75.8 | 0.70 | 2.65 | 99.4 | 0.66 | 2.97 | 99.6 | 0.67 | 0.50 | 99.8 | 0.66 | 5.74 | 100.0 | 0.65 | 0.62 | 97.2 | 0.72 | 1.55 | 96.8 | 0.71 | 0.86 |
| 0.5–0.6 | 99.6 | 0.67 | 1.33 | 74.2 | 0.71 | 2.52 | 99.6 | 0.66 | 2.61 | 100.0 | 0.66 | 0.51 | 99.4 | 0.68 | 4.98 | 100.0 | 0.64 | 0.62 | 96.8 | 0.73 | 1.65 | 93.8 | 0.69 | 0.65 |
| 0.6–0.7 | 99.8 | 0.67 | 1.34 | 88.2 | 0.68 | 2.03 | 99.6 | 0.66 | 2.72 | 100.0 | 0.68 | 0.47 | 99.6 | 0.67 | 4.89 | 99.8 | 0.67 | 0.56 | 96.0 | 0.71 | 1.54 | 93.0 | 0.70 | 0.78 |
| 0.7–0.8 | 99.8 | 0.66 | 1.36 | 80.2 | 0.71 | 1.72 | 99.8 | 0.68 | 2.19 | 99.6 | 0.66 | 0.55 | 99.8 | 0.67 | 5.31 | 99.8 | 0.67 | 0.52 | 95.8 | 0.70 | 1.63 | 94.4 | 0.70 | 0.64 |
| 0.8–0.9 | 99.8 | 0.68 | 1.27 | 80.2 | 0.72 | 1.50 | 99.4 | 0.69 | 2.07 | 100.0 | 0.67 | 0.50 | 99.4 | 0.68 | 4.69 | 99.4 | 0.67 | 0.59 | 96.8 | 0.73 | 1.30 | 95.6 | 0.69 | 0.68 |

As shown in Figure 9, we condition the data by six similarity bins used during construction and plot the ECFP4 Tanimoto similarity between optimized and source molecules. When moving from the 0.3–0.4 bin up to 0.5–0.6, the distribution's dominant mode shifts rightward and the low-similarity tail compresses, indicating improved scaffold fidelity under tighter filtering. Increasing the threshold further from 0.6 to 0.9 leaves the peak location nearly unchanged but sharpens the peak (mass concentrates around it), revealing clear diminishing returns beyond 0.6. This pattern aligns with the table-level results: similarity improves with the threshold, yet gains are marginal past 0.5–0.6. Together with the observed decrease in RI at higher thresholds, the figure supports selecting a 0.5–0.6 threshold as a balanced operating point between scaffold preservation and improvability.

Table 14: Appendix single-property results (Part I): BBBP, QED, pLogP, and HIA. Results are reported as mean ± std over runs. Best mean values in each column are bold.

| Model | BBBP | | | QED | | | pLogP | | | HIA | | |
|---|---|---|---|---|---|---|---|---|---|---|---|---|
| | SR ↑ | RI ↑ | SIM ↑ | SR ↑ | RI ↑ | SIM ↑ | SR ↑ | RI ↑ | SIM ↑ | SR ↑ | RI ↑ | SIM ↑ |
| SFT-LLM$_{Mistral}$ | 99.55 ± 0.10 | 1.755 ± 0.013 | 0.560 ± 0.000 | **100.00 ± 0.00** | 0.810 ± 0.000 | **0.570 ± 0.000** | **99.95 ± 0.10** | 9.808 ± 0.076 | **0.570 ± 0.000** | 98.95 ± 0.10 | 5.048 ± 0.072 | **0.560 ± 0.000** |
| DPO-LLM$_{Mistral}$ | 99.533 ± 0.115 | 1.973 ± 0.015 | 0.520 ± 0.010 | 99.533 ± 0.115 | 0.920 ± 0.000 | 0.533 ± 0.006 | 99.60 ± 0.20 | 17.427 ± 0.051 | 0.510 ± 0.065 | 97.333 ± 0.416 | 6.740 ± 0.207 | 0.493 ± 0.006 |
| SFT-LLM$_{LLaMA}$ | **99.733 ± 0.115** | 1.757 ± 0.006 | **0.577 ± 0.006** | 99.80 ± 0.00 | 0.860 ± 0.000 | **0.570 ± 0.000** | 99.933 ± 0.115 | 9.017 ± 0.047 | **0.570 ± 0.000** | 98.467 ± 0.306 | 5.687 ± 0.093 | **0.560 ± 0.000** |
| DPO-LLM$_{LLaMA}$ | 99.533 ± 0.115 | **1.987 ± 0.047** | 0.510 ± 0.006 | 99.80 ± 0.503 | **0.977 ± 0.015** | 0.543 ± 0.006 | 99.467 ± 0.260 | **18.097 ± 0.027** | 0.480 ± 0.006 | **99.467 ± 0.115** | **7.460 ± 0.166** | 0.520 ± 0.006 |

Table 15: Appendix single-property results (Part II): DRD2, Mutag, GSK3$\beta$, and JNK3. Results are reported as mean ± std over runs. Best mean values in each column are bold.

| Model | DRD2 | | | Mutag | | | GSK3$\beta$ | | | JNK3 | | |
|---|---|---|---|---|---|---|---|---|---|---|---|---|
| | SR ↑ | RI ↑ | SIM ↑ | SR ↑ | RI ↑ | SIM ↑ | SR ↑ | RI ↑ | SIM ↑ | SR ↑ | RI ↑ | SIM ↑ |
| SFT-LLM$_{Mistral}$ | **98.10 ± 0.258** | 8.655 ± 0.142 | 0.570 ± 0.000 | **99.80 ± 0.00** | 0.675 ± 0.006 | **0.570 ± 0.000** | **98.80 ± 0.033** | 10.256 ± 0.153 | **0.590 ± 0.000** | 98.20 ± 0.10 | 3.940 ± 0.024 | **0.580 ± 0.000** |
| DPO-LLM$_{Mistral}$ | 95.60 ± 0.529 | **14.820 ± 0.108** | 0.547 ± 0.006 | 99.40 ± 0.20 | **0.853 ± 0.006** | 0.520 ± 0.016 | 98.25 ± 0.486 | 17.116 ± 0.982 | 0.520 ± 0.000 | **98.90 ± 0.166** | **8.590 ± 0.049** | 0.490 ± 0.006 |
| SFT-LLM$_{LLaMA}$ | 96.60 ± 0.529 | 7.760 ± 0.313 | **0.583 ± 0.006** | 99.533 ± 0.115 | 0.670 ± 0.008 | 0.557 ± 0.006 | 98.60 ± 0.447 | 12.790 ± 0.046 | 0.510 ± 0.005 | 98.60 ± 0.156 | 5.770 ± 0.066 | 0.550 ± 0.000 |
| DPO-LLM$_{LLaMA}$ | 97.80 ± 1.858 | 10.630 ± 0.154 | 0.550 ± 0.012 | 99.00 ± 0.00 | 0.780 ± 0.006 | 0.520 ± 0.010 | 98.00 ± 0.146 | **20.010 ± 0.650** | 0.490 ± 0.002 | 98.20 ± 0.450 | 7.810 ± 0.136 | 0.470 ± 0.005 |

### C.6.2 Property Gap Threshold

With the similarity window fixed at $[0.7, 0.8]$, we examine how the property-difference bin affects performance. For eight single-property tasks (BBBP, DRD2, HIA, Mutag, pLogP, QED, GSK3$\beta$, JNK3), we sample 2000 molecular pairs per bin $[0, 0.1)$, $[0.1, 0.2)$, ..., $[0.8, 0.9)$. Sampling is calibrated so that the distribution of property differences is matched across bins, isolating the effect of the property threshold. Results are summarized in Table 13.

Under the fixed similarity constraint, SIM varies only marginally across bins, confirming effective control of similarity. SR is mostly stable but endpoint-dependent: most tasks remain at high SR, while DRD2 is more sensitive to the gap (higher at moderate gaps, declining at the largest); GSK3$\beta$ and JNK3 show mild downward trends. The dominant pattern is a negative association between RI and the property gap: as the gap increases, RI consistently decreases for most endpoints (with BBBP and Mutag changing little). Because the distributions are matched across bins, this indicates that very large gaps compress the learnable local edit space—pushing the problem toward near single-site modifications—thereby reducing achievable relative improvement and, for some endpoints, also depressing SR. Under fixed similarity, small-to-moderate property thresholds are advisable to balance improvability (higher RI/SR) against scaffold preservation and robustness.

### C.6.3 Single property optimization

From Table 3, DPO exhibits comparable SR to SFT: for most single-property tasks, SFT-LLM and DPO-LLM both reach near-maximum or 100% SR, indicating that single-property optimization is relatively easy once the model is SFT-aligned. However, DPO consistently improves RI over SFT (e.g., for pLogP, QED, DRD2, and JNK3/GSK3$\beta$), suggesting that DPO better captures the preference that small structural edits lead to larger property gains. At the same time, SIM is slightly lower for DPO than for SFT, reflecting broader structural exploration to achieve higher property scores. This RI–SIM trade-off is common across tasks; the DPO SIM values remain within an acceptable range, and we further analyze the interaction between similarity constraints, success rate, and relative improvement in a later section.

Figure 10 shows kernel densities of the property difference for single-property tasks. Curves correspond to the training-pair baseline, SFT, and DPO. For most endpoints, SFT and DPO shift the mass to the right relative to the baseline, indicating positive improvement; for Mutag (to be reduced), the distribution shifts left. DPO (green/blue) typically exhibits a stronger right shift and heavier right tail than SFT (orange/red), reflecting larger and more consistent gains.

Closed-source (ChatGPT-4o, Claude-3.5) and open-source (LLaMA, Mistral) general LLMs under zero/five-shot underperform SFT/DPO-tuned models. In the zero-shot setting, ChatGPT-4o generally exceeds Claude-3.5 in SR/RI yet still lags behind tuned models; notably, Claude-3.5 shows a very low SR on BBBP (17.6), indicating that out-of-the-box general LLMs struggle to transfer to specialized chemical optimization. In the five-shot setting, general LLMs improve SR and RI markedly (e.g., LLaMA, GPT-4o, and Claude across several tasks) but substantially reduce SIM (e.g., LLaMA five-shot SIM $\approx$ 0.15–0.27 on multiple tasks),

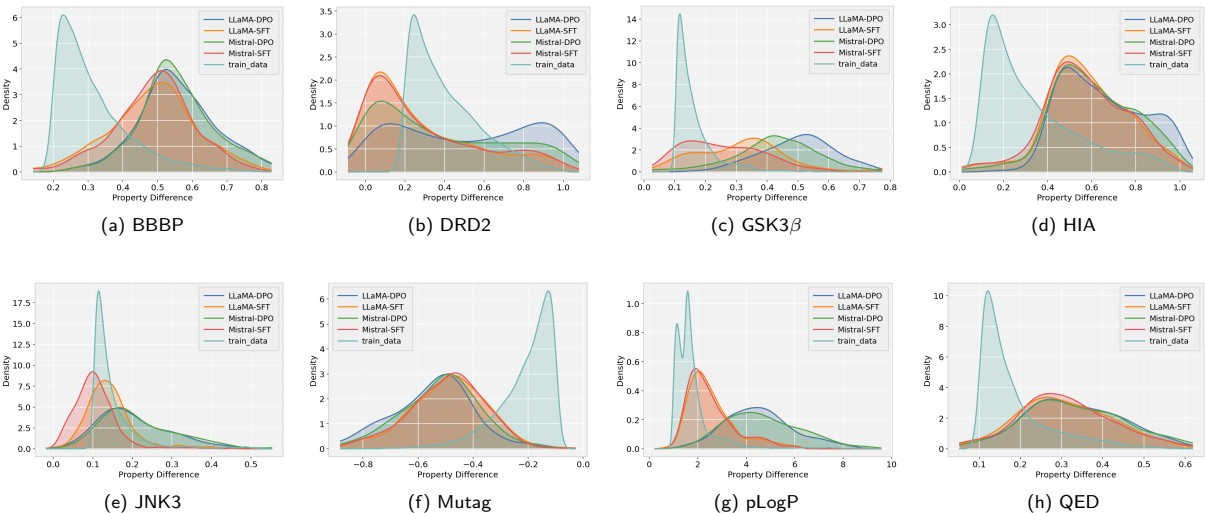

Figure 10: Density of Property Differences Between Source and Optimized Molecules Across Tasks. The figure shows kernel density distributions of the property difference for each task. Subfig (f) is a reduction-oriented task, so its distribution trends in the opposite direction (toward negative values).

implying that example-based improvement often comes at the cost of scaffold preservation. While closed-source models sometimes perform better on more generic properties (e.g., QED, pLogP) due to stronger background knowledge, their advantage is less consistent on more complex models.

ChemLLM tends to maintain higher SIM but exhibits relatively low SR/RI, highlighting that textual chemistry knowledge does not automatically translate into controllable molecular optimization. $LlaSmol_{Mistral}$ achieves high SR with moderate RI but lower SIM (e.g., BBBP SIM = 0.25), echoing the success vs. similarity trade-off.

### C.6.4 Multi property optimization

We evaluate the proposed DPO under more challenging multi-property settings. From Table 4, all models degrade relative to single-property tasks, yet DPO consistently improves SR and RI over SFT with acceptable decreases in SIM reflecting broader exploration for higher property gains and how different the worse molecule with the better molecule.

Mistral backbone. Compared to SFT, DPO increases SR across all five combinations: BDP +0.3 (81.6→81.93), BDQ +1.92 (83.95→85.87), DPQ +7.28 (62.85→70.13), and BDPQ +8.13 (57→65.13). RI also improves on every task, while SIM decreases slightly (e.g., BDP 0.55→0.48), exhibiting the typical RI–SIM trade-off.

LLaMA backbone. DPO again outperforms SFT on SR across the board: BDP +5.47 (78.6→84.07), BDQ +7.72 (78.95→86.67), BPQ +2.58 (94.55→97.13), DPQ +10.73 (57.6→68.33), and BDPQ +13.05 (46.15→59.2). RI increases, while SIM decreases modestly (e.g., BDP 0.50→0.47). Notably, the largest SR/RI gains appear in the most difficult setting (BDPQ) for both backbones.

General and specialized baselines. General-purpose LLMs (LLaMA, Mistral, GPT-4o, Claude-3.5) show low SR in multi-property tasks under zero/five-shot. Five-shot improves SR/RI but substantially reduces SIM (e.g., LLaMA five-shot DPQ SIM=0.44; Claude-3.5 five-shot DPQ SIM=0.37), deviating from scaffold preservation. Among specialized models, ChemLLM attains very low SR (often single-digit or zero), while $LlaSmol_{Mistral}$ reaches high SR (e.g., BPQ 86.0) with moderate RI and lower SIM. Overall, DPO delivers more pronounced and robust gains than SFT in multi-property scenarios, especially on BDPQ, achieving higher SR and RI with a reasonable similarity trade-off.

Table 16: Multi-property results in the appendix (Part I). Results are reported as mean ± std over runs. Best mean values in each column are bold.

| Model | BDP | | | BDQ | | | BPQ | | |
|---|---|---|---|---|---|---|---|---|---|
| | SR ↑ | SIM ↑ | RI ↑ | SR ↑ | SIM ↑ | RI ↑ | SR ↑ | SIM ↑ | RI ↑ |
| SFT-LLM$_{Mistral}$ | 81.60 ± 0.67 | **0.560 ± 0.000** | 3.87 ± 0.05 | 83.95 ± 0.77 | 0.550 ± 0.000 | 4.98 ± 0.07 | 97.05 ± 0.34 | **0.538 ± 0.005** | 1.45 ± 0.01 |
| DPO-LLM$_{Mistral}$ | 81.93 ± 0.50 | 0.497 ± 0.006 | 5.15 ± 0.19 | 85.87 ± 0.50 | 0.517 ± 0.006 | 6.22 ± 0.06 | 97.07 ± 0.12 | 0.467 ± 0.006 | 1.74 ± 0.03 |
| SFT-LLM$_{LLaMA}$ | 78.60 ± 0.59 | 0.558 ± 0.005 | 3.64 ± 0.10 | 78.95 ± 1.20 | **0.563 ± 0.005** | 4.49 ± 0.09 | 94.55 ± 0.47 | 0.495 ± 0.006 | 1.55 ± 0.02 |
| DPO-LLM$_{LLaMA}$ | **84.07 ± 0.44** | 0.470 ± 0.010 | **6.67 ± 0.33** | **86.67 ± 1.70** | 0.510 ± 0.000 | **6.52 ± 0.04** | **97.13 ± 0.99** | 0.450 ± 0.010 | **1.98 ± 0.06** |

Table 17: Multi-property results in the appendix (Part II). Results are reported as mean ± std over runs. Best mean values in each column are bold.

| Model | DPQ | | | BDPQ | | |
|---|---|---|---|---|---|---|
| | SR ↑ | SIM ↑ | RI ↑ | SR ↑ | SIM ↑ | RI ↑ |
| SFT-LLM$_{Mistral}$ | 62.85 ± 1.91 | **0.513 ± 0.005** | 2.42 ± 0.11 | 57.00 ± 1.49 | **0.523 ± 0.010** | 3.05 ± 0.07 |
| DPO-LLM$_{Mistral}$ | **70.13 ± 0.42** | 0.443 ± 0.006 | **4.30 ± 0.19** | **65.13 ± 1.36** | 0.430 ± 0.010 | 5.65 ± 0.18 |
| SFT-LLM$_{LLaMA}$ | 57.60 ± 0.94 | 0.510 ± 0.006 | 2.12 ± 0.09 | 46.15 ± 0.87 | 0.498 ± 0.010 | 3.36 ± 0.11 |
| DPO-LLM$_{LLaMA}$ | 68.33 ± 1.94 | 0.460 ± 0.010 | 3.39 ± 0.19 | 59.20 ± 1.80 | 0.420 ± 0.010 | **5.97 ± 0.41** |

### C.6.5 Parameter Sensitivity Results

This experiment evaluates the impact of DPO hyperparameter choices on model optimization performance and, based on the results, selects suitable hyperparameters for subsequent training. The result is shown in Table 5

In the multi-property setting, we systematically sweep two key DPO hyperparameters—learning rate and $\beta$. The learning rate governs how strongly the model adapts to triplet preferences, while $\beta$ scales the log-probability gap between "better" and "worse" samples, acting as a gain on the preference signal. The results exhibit a consistent pattern: increasing the learning rate generally raises success rate (SR) and relative improvement (RI) while reducing similarity (SIM); increasing $\beta$ weakens sensitivity to preference gaps, leading to lower SR/RI and a modest recovery in SIM, with diminishing returns at the larger end. Taken together, these two knobs trace a performance–similarity trade-off: stronger preference alignment yields larger property gains at the cost of similarity.

Accordingly, we recommend choosing operating points by application goals. For improvement-oriented scenarios, use a relatively higher learning rate with a smaller $\beta$ to reinforce the mapping from local edits to property gains. For scaffold-preserving scenarios, favor a lower learning rate with a larger $\beta$ to obtain more conservative edits and higher SIM. This aligns with our earlier observation of a negative correlation between RI and SIM and provides practical guidance for hyperparameter tuning and data filtering in DPO-based molecular optimization.

As shown in Figure 6, this panel visualizes parameter sensitivity on the BPQ task.. Panels (a–d) use decreasing learning rates; within each panel, the three rows correspond to $\beta$=0.1 (top), 0.3 (middle), and 0.5 (bottom). As the learning rate decreases, the extent of edits shrinks and property gains diminish; at a fixed learning rate, larger $\beta$ further suppresses the edit magnitude and slightly increases similarity.

### C.6.6 Analysis on single property task search budget.

To complement the budget-matched comparison in the main paper, we further report the performance of non-LLM baselines under larger search budgets in Table 18. This analysis is included for two reasons. First, methods such as GRAPHGA, JTVAE, and REINVENT are search-based or iterative generators whose performance may continue to improve when given more optimization steps. Second, reporting multiple budgets helps clarify whether the gap observed in the main text is merely due to limited search budget, or reflects a more fundamental difference in optimization behavior.

The results show that increasing the budget often leads to substantial gains in RI, and sometimes also improves SR. However, these gains do not translate into correspondingly high similarity to the source molecule: SIM remains consistently low across methods and budgets, and in some cases even decreases as the budget

Table 18: Budget sensitivity of non-LLM baselines on single-property optimization tasks under the PMO framework.

| Method | Budget | DRD2 | | | GSK3$\beta$ | | | pLogP | | | QED | | |
|---|---|---|---|---|---|---|---|---|---|---|---|---|---|
| | | SR↑ | SIM↑ | RI↑ | SR↑ | SIM↑ | RI↑ | SR↑ | SIM↑ | RI↑ | SR↑ | SIM↑ | RI↑ |
| GraphGA | 20 | 30.80 | 0.210 | 12.86 | 100.00 | 0.077 | 15.48 | 100.00 | 0.154 | 11.00 | 100.00 | 0.159 | 1.20 |
| | 50 | 69.00 | 0.187 | 84.70 | 100.00 | 0.077 | 15.48 | 100.00 | 0.113 | 16.18 | 100.00 | 0.164 | 1.34 |
| | 100 | 93.80 | 0.138 | 324.20 | 100.00 | 0.078 | 15.39 | 100.00 | 0.113 | 16.18 | 100.00 | 0.191 | 1.32 |
| | 200 | 93.40 | 0.142 | 453.98 | 100.00 | 0.091 | 15.63 | 100.00 | 0.112 | 19.90 | 100.00 | 0.199 | 1.31 |
| JTVAE | 20 | 99.60 | 0.111 | 477.19 | 84.60 | 0.122 | 5.86 | 100.00 | 0.124 | 15.79 | 99.80 | 0.151 | 1.16 |
| | 50 | 98.80 | 0.114 | 499.34 | 87.95 | 0.124 | 6.34 | 100.00 | 0.124 | 15.79 | 100.00 | 0.163 | 1.23 |
| | 100 | 98.80 | 0.120 | 580.30 | 93.97 | 0.125 | 6.68 | 100.00 | 0.124 | 15.79 | 100.00 | 0.176 | 1.28 |
| | 200 | 98.60 | 0.125 | 639.35 | 97.10 | 0.124 | 7.36 | 100.00 | 0.111 | 17.53 | 100.00 | 0.189 | 1.30 |
| REINVENT | 20 | 52.00 | 0.188 | 38.41 | 57.59 | 0.165 | 0.81 | 100.00 | 0.105 | 19.06 | 100.00 | 0.138 | 1.32 |
| | 50 | 97.40 | 0.144 | 345.70 | 88.39 | 0.140 | 3.85 | 100.00 | 0.105 | 19.06 | 100.00 | 0.158 | 1.28 |
| | 100 | 95.20 | 0.134 | 599.35 | 94.42 | 0.140 | 5.55 | 100.00 | 0.108 | 19.73 | 100.00 | 0.177 | 1.27 |
| | 200 | 95.20 | 0.136 | 743.29 | 96.43 | 0.132 | 8.27 | 100.00 | 0.111 | 20.06 | 100.00 | 0.193 | 1.29 |

Table 19: Budget sensitivity of non-LLM baselines on three-property optimization tasks under the PMO framework. Entries with no successful molecules have undefined SIM/RI and are marked as −.

| Method | Budget | DRD2+pLogP+QED | | | GSK3$\beta\beta$+DRD2+pLogP | | | GSK3$\beta\beta$+QED+DRD2 | | | GSK3$\beta$+QED+pLogP | | |
|---|---|---|---|---|---|---|---|---|---|---|---|---|---|
| | | SR↑ | SIM↑ | RI↑ | SR↑ | SIM↑ | RI↑ | SR↑ | SIM↑ | RI↑ | SR↑ | SIM↑ | RI↑ |
| GraphGA | 20 | 9.0 | 0.154 | 3.13 | 19.9 | 0.144 | 11.04 | 5.9 | 0.101 | 6.30 | 4.1 | 0.153 | 0.60 |
| | 50 | 2.6 | 0.128 | 1.57 | 0.0 | − | − | 9.3 | 0.133 | 8.76 | 0.0 | − | − |
| | 100 | 2.6 | 0.128 | 1.57 | 0.0 | − | − | 78.8 | 0.098 | 45.50 | 0.0 | − | − |
| | 200 | 26.2 | 0.099 | 3.15 | 41.6 | 0.110 | 32.18 | 79.3 | 0.095 | 63.65 | 40.1 | 0.097 | 1.93 |
| JTVAE | 20 | 51.0 | 0.101 | 2.44 | 69.3 | 0.115 | 8.02 | 30.7 | 0.101 | 13.82 | 91.8 | 0.094 | 1.79 |
| | 50 | 51.0 | 0.101 | 2.44 | 69.3 | 0.115 | 8.02 | 35.7 | 0.107 | 29.64 | 91.8 | 0.094 | 1.79 |
| | 100 | 51.0 | 0.101 | 2.44 | 69.3 | 0.115 | 8.02 | 39.3 | 0.105 | 45.81 | 91.8 | 0.094 | 1.79 |
| | 200 | 0.4 | 0.098 | 1.17 | 0.2 | 0.075 | 2.98 | 34.9 | 0.106 | 66.96 | 0.3 | 0.094 | 2.71 |
| REINVENT | 20 | 35.8 | 0.112 | 2.05 | 4.8 | 0.137 | 7.86 | 0.8 | 0.135 | 1.47 | 85.4 | 0.109 | 1.69 |
| | 50 | 17.2 | 0.141 | 2.08 | 4.8 | 0.137 | 7.86 | 72.1 | 0.105 | 49.62 | 84.1 | 0.102 | 2.17 |
| | 100 | 13.4 | 0.096 | 1.93 | 0.2 | 0.220 | 6.25 | 96.6 | 0.076 | 153.25 | 0.0 | − | − |
| | 200 | 12.0 | 0.102 | 1.89 | 16.2 | 0.115 | 4.89 | 96.9 | 0.070 | 155.80 | 0.5 | 0.068 | 0.63 |

grows. This suggests that additional search budget is mainly spent on exploring molecules farther away from the source structure, rather than discovering better local edits around the same scaffold. Therefore, while non-LLM baselines remain competitive when larger structural drift is allowed, the results further support our main claim that LLMs are better suited for source-conditioned, scaffold-preserving molecular optimization.

### C.6.7 Analysis on multi task search budget.

Table 19 provides additional results for non-LLM baselines under larger search budgets on the three-property optimization tasks. This analysis is useful because methods such as GRAPHGA, JTVAE, and REINVENT are search-based or iterative generators, and their performance may change substantially when more optimization steps are allowed.

Overall, increasing the budget sometimes improves RI, and in some cases also improves SR. However, these gains rarely translate into high structural similarity. Across most tasks and methods, SIM remains in a very low range even when RI grows substantially, suggesting that additional search steps are mainly used to explore molecules that drift farther away from the source scaffold rather than to discover better local edits. This trend is especially visible on tasks such as GSK3$\beta$+QED+DRD2, where the RI of several non-LLM baselines increases dramatically at higher budgets while similarity remains around 0.07–0.11.

Another notable observation is that multi-property optimization with non-LLM baselines can be unstable across budgets. For some method–task pairs, no successful molecules are obtained at certain budgets, leading

Table 20: Budget sensitivity of GenMol on single-property optimization tasks.

| Method | Budget | DRD2 | | | GSK3$\beta$ | | | pLogP | | | QED | | |
|---|---|---|---|---|---|---|---|---|---|---|---|---|---|
| | | SR↑ | SIM↑ | RI↑ | SR↑ | SIM↑ | RI↑ | SR↑ | SIM↑ | RI↑ | SR↑ | SIM↑ | RI↑ |
| GenMol | 20 | 10.80 | 0.448 | 1.54 | 53.35 | 0.449 | 2.99 | 58.00 | 0.442 | 4.16 | 62.80 | 0.427 | 0.75 |
| GenMol | 50 | 19.80 | 0.403 | 2.86 | 72.10 | 0.395 | 4.80 | 71.60 | 0.409 | 7.81 | 72.80 | 0.404 | 0.87 |
| GenMol | 100 | 28.80 | 0.379 | 4.77 | 75.22 | 0.370 | 6.26 | 74.80 | 0.389 | 10.46 | 73.60 | 0.399 | 0.90 |
| GenMol | 200 | 34.20 | 0.369 | 6.10 | 76.34 | 0.362 | 7.24 | 75.80 | 0.381 | 13.67 | 75.00 | 0.392 | 0.92 |

Table 21: Budget sensitivity of GenMol on three-property optimization tasks.

| Method | Budget | DPQ | | | GDP | | | GQD | | | GQP | | |
|---|---|---|---|---|---|---|---|---|---|---|---|---|---|
| | | SR↑ | SIM↑ | RI↑ | SR↑ | SIM↑ | RI↑ | SR↑ | SIM↑ | RI↑ | SR↑ | SIM↑ | RI↑ |
| GenMol | 20 | 0.60 | 0.443 | 0.57 | 0.23 | 0.426 | 5.56 | 0.00 | 0.409 | 4.97 | 2.47 | 0.417 | 0.67 |
| GenMol | 50 | 0.60 | 0.411 | 0.87 | 0.92 | 0.394 | 10.89 | 0.00 | 0.380 | 5.55 | 4.12 | 0.389 | 0.84 |
| GenMol | 100 | 1.60 | 0.398 | 1.15 | 0.69 | 0.373 | 13.29 | 0.00 | 0.374 | 5.67 | 4.40 | 0.383 | 0.94 |
| GenMol | 200 | 2.20 | 0.395 | 1.50 | 0.69 | 0.363 | 16.38 | 0.00 | 0.372 | 5.79 | 4.40 | 0.380 | 1.10 |

to undefined SIM/RI values (marked as – in Table 19). This further suggests that larger search budgets do not reliably solve the challenge of high-order scaffold-preserving optimization.

Taken together, these results indicate that the relative weakness of non-LLM baselines in the main text is not simply a budget issue. Even when additional optimization steps are provided, their improvements tend to come from lower-similarity exploration rather than better source-conditioned local refinement. This is consistent with our main conclusion that LLMs are better aligned with the goal of scaffold-preserving, source-conditioned multi-property optimization.

### C.6.8 GenMol budget sensitivity

Across both single-property and three-property tasks, increasing the GenMol sampling budget generally improves SR and RI, but the average similarity decreases. This behavior is consistent with broader chemical-space exploration under larger budgets. In contrast, SCPT is designed to operate as a source-conditioned local editor, where the objective is not only to improve property scores but also to remain within a scaffold-preserving neighborhood.

## D   Failure Analysis

### D.1   Failure Categorization Protocol

For each failed test case, we first select the best generated candidate using the same selection rule as in the main evaluation. We then assign the failure to one of three categories. Invalid indicates that all generated candidates are unparsable or cannot be evaluated by the property oracle. Wrong Direction indicates that at least one target property changes in the opposite direction. Insufficient Improvement indicates that all target directions are correct but the improvement does not reach the task-specific threshold.

### D.2   Task-level failure breakdown

Table 22 and Table 23 report the failure composition for single-property and multi-property tasks, respectively. The percentages of Invalid, Wrong Direction, and Insufficient Improvement are computed among failed cases within each task, and therefore sum to 100% for each row.

Single-property failures are dominated by invalid generations, whereas multi-property failures are dominated by wrong-direction or insufficient-improvement cases. This supports the main-text conclusion that multi-objective optimization changes the failure mode from molecular validity to simultaneous directional control.

Table 22: Failure composition on single-property tasks. The last three columns denote percentages among failed cases, not percentages among all test cases.

| Task | Avg. SR | Invalid | Wrong Direction | Insufficient Improvement |
|------|---------|---------|-----------------|--------------------------|
| pLogP | 94.9 | 94.9 | 5.1 | 0.0 |
| BBBP | 94.4 | 85.4 | 14.6 | 0.0 |
| QED | 93.5 | 68.2 | 31.8 | 0.0 |
| HIA | 92.8 | 56.1 | 5.4 | 38.5 |
| Mutagenicity | 94.1 | 56.2 | 20.1 | 23.7 |
| DRD2 | 87.5 | 43.3 | 56.7 | 0.0 |

Table 23: Failure composition on multi-property tasks. The last three columns denote percentages among failed cases, not percentages among all test cases.

| Task | Avg. SR | Invalid | Wrong Direction | Insufficient Improvement |
|------|---------|---------|-----------------|--------------------------|
| BBBP+pLogP+QED | 91.4 | 23.2 | 31.3 | 45.5 |
| BBBP+DRD2+pLogP | 74.7 | 21.9 | 78.1 | 0.0 |
| BBBP+DRD2+QED | 73.9 | 20.4 | 79.6 | 0.0 |
| DRD2+pLogP+QED | 57.9 | 15.6 | 84.4 | 0.0 |
| BBBP+DRD2+pLogP+QED | 49.9 | 11.1 | 80.1 | 8.8 |

### D.3 Model-level failure breakdown

DPO tends to produce more aggressive edits and therefore has a higher invalid-generation share among failures, whereas SFT is more conservative but its failures are dominated by wrong-direction cases. This is consistent with the main trade-off observed in RQ3: stronger preference updates improve gains but may reduce structural conservativeness.

### D.4 Endpoint-specific distribution shift

These results indicate that failure rates increase when the source molecules lie outside the property ranges frequently observed during training. This effect is especially visible for endpoints with strong source-property distribution shift.

## E   Scaffold Distribution and Generalization

### E.1 Scaffold extraction and counting protocol

We extract Bemis–Murcko scaffolds from the SCPT-filtered training set and count the number of training examples contributed by each scaffold. This analysis characterizes training-distribution concentration after filtering. It is distinct from train-test leakage control: excluding test molecules from training evaluates generalization, whereas scaffold imbalance measures how unevenly filtered supervision is distributed across scaffold groups.

### E.2 Overall scaffold concentration

The filtered training data exhibit a long-tailed scaffold distribution: many scaffolds contribute only a few examples, while a small subset of frequent scaffolds contributes a substantial fraction of the filtered supervision.

Table 24: Model-level failure composition.

| Model group | Avg. SR | Invalid | Wrong Direction | Insufficient Improvement |
|---|---|---|---|---|
| DPO | 75.5 | 41.2 | 48.3 | 10.5 |
| SFT | 87.0 | 2.9 | 97.1 | 0.0 |
| LLaMA | 73.9 | 35.1 | 55.7 | 9.2 |
| Mistral | 89.2 | 5.7 | 94.3 | 0.0 |

Table 25: Endpoint-specific source-property distribution shift.

| Property | Train source mean | Test source mean | Shift | Test below train P25 |
|---|---|---|---|---|
| DRD2 | 0.145 | 0.007 | -0.138 | 92.0 |
| BBBP | 0.511 | 0.355 | -0.156 | 56.8 |
| HIA | 0.576 | 0.316 | -0.260 | 55.8 |
| QED | 0.612 | 0.470 | -0.142 | 48.2 |
| pLogP | -0.963 | -0.998 | -0.035 | 25.2 |
| Mutagenicity | 0.468 | 0.664 | +0.196 | 0.0 |

### E.3  Per-task scaffold concentration

The filtered data are long-tailed across scaffolds. This concentration is more consequential in multi-property tasks, where the absolute number of valid triplets is smaller and a few scaffolds may contribute a larger fraction of the supervision.

### E.4  Scaffold-level generalization under strict train-test separation

The following analysis evaluates scaffold-level generalization under the strict train-test separation used in our experiments. Low train-test scaffold overlap should not be interpreted as data leakage; rather, it reflects a deliberately strict evaluation protocol.

The low overlap reflects a strict generalization protocol rather than data leakage. It also indicates that the model is often evaluated on scaffold regions not directly observed during SCPT training.

Unseen scaffolds generally have higher failure rates, suggesting that scaffold-level distribution shift contributes to residual failures. This motivates future work on scaffold-balanced sampling, scaffold-aware reweighting, and active data acquisition.

Table 26: Overall scaffold concentration after SCPT filtering. Percentage values are reported in %.

| Statistic | Value |
|---|---|
| Training examples | 828,097 |
| Unique scaffolds | 66,689 |
| Maximum examples for one scaffold | 11,974 |
| Median examples per scaffold | 4 |
| Singleton scaffolds | 10,436 |
| Singleton scaffold share | 15.6 |
| Data contributed by singleton scaffolds | 1.3 |
| Top 10 scaffolds contribution | 5.2 |
| Top 500 scaffolds contribution | 31.9 |
| Top 1000 scaffolds contribution | 40.5 |

Table 27: Per-task scaffold concentration after SCPT filtering. Share values are reported in %.

| Task | Gini | Top 10 scaffold share | Singleton scaffold share |
|---|---|---|---|
| DRD2 | 0.708 | 5.3 | 47.6 |
| QED | 0.633 | 5.0 | 67.0 |
| Mutagenicity | 0.624 | 5.2 | 66.0 |
| BBBP+DRD2+QED | 0.582 | 12.4 | 52.0 |
| BBBP | 0.553 | 7.0 | 71.6 |
| BBBP+DRD2+pLogP+QED | 0.503 | 30.8 | 62.6 |

Table 28: Train-test scaffold overlap.

| Task | Train scaffolds | Test scaffolds | Overlap | Overlap rate |
|---|---|---|---|---|
| BBBP+DRD2+pLogP+QED | 281 | 377 | 3 | 0.8 |
| BBBP+DRD2+QED | 1,421 | 391 | 2 | 0.5 |
| BBBP+DRD2+pLogP | 807 | 452 | 6 | 1.3 |
| DRD2+pLogP+QED | 799 | 414 | 6 | 1.4 |
| BBBP+pLogP+QED | 2,396 | 416 | 19 | 4.6 |
| BBBP | 16,443 | 445 | 34 | 7.6 |
| Mutagenicity | 33,586 | 473 | 69 | 14.6 |

Table 29: Seen vs. unseen scaffold failure rate.

| Task | Seen-scaffold failure rate | Unseen-scaffold failure rate | Ratio |
|---|---|---|---|
| pLogP | 11.4 | 23.0 | 2.01× |
| BBBP | 15.2 | 22.9 | 1.51× |
| BBBP+pLogP+QED | 20.4 | 29.4 | 1.44× |
| DRD2+pLogP+QED | 38.5 | 59.8 | 1.55× |
| HIA | 20.1 | 27.7 | 1.38× |

