# OpenReview forum: "Scaffold-Conditioned Preference Triplets for Controllable Molecular Optimization with Large Language Models"
_TMLR — Decision pending for TMLR_

### Review · Reviewer_14FM · 2026-05-03

**Summary Of Contributions:**

This paper seeks to improve LLM based molecular optimization by introducing "Scaffolding-Condition Preference Triplets." The purported advantages of the proposed method are to produce a systematic chemistry pipeline with a pretrained LLM for initialization. The authors then conduct extensive experiments on the role of locality in the data comparing against both LLMs and more traditional baselines.

**Audience:**

Yes

**Audience Explanation:**

Molecular optimization is an important scientific problem of wide interest

**Claims And Evidence:**

Yes

**Claims Explanation:**

Extensive experiments

**Requested Changes:**

As someone who is not an expert on either LLMs or on chemistry, I found it hard to understand the motivation and the novelty of the proposed method. I suggest that the authors revise the paper to more clearly explain what scaffolding is in the context of molecular optimization, why it is important, and how the proposed method compares, from an algorithmic perspective, to standard methods.

---

> ### Author Response · Authors · 2026-05-07
> **Clarifying Scaffold Motivation and Algorithmic Novelty**
>
> ## Response to Reviewer
>
> We thank the reviewer for the positive assessment of our experimental evidence and for raising this important clarity issue. We agree that, especially for readers who are not specialists in either LLMs or chemistry, the motivation and novelty of scaffold-conditioned molecular optimization should be explained more explicitly and in a more self-contained way.
>
> We would also like to clarify that the current manuscript does address these points, although they may not be sufficiently consolidated. In the Introduction, we motivate molecular optimization as a lead-optimization problem, where chemists usually start from a validated lead compound and perform local modifications while preserving key structural motifs. In Section 3.1, we formulate the optimization objective with an explicit similarity/scaffold-preservation constraint. In Section 3.2, we describe how SCPT constructs scaffold-conditioned preference triplets. Finally, in Section 4.5, we compare our method with representative non-LLM molecular optimization baselines, including GraphGA, JTVAE, and REINVENT, under a similarity-aware objective. These experiments show that our method is not merely improving property scores, but is particularly effective in preserving structural locality while optimizing molecular properties.
>
> To directly address the reviewer’s question, in the context of molecular optimization, a *scaffold* refers to the core structural framework of a molecule, typically including the main ring systems and linker structures that support peripheral substituents. In lead optimization, this scaffold is important because it often carries key biological activity, synthetic feasibility, and interpretability. Therefore, a useful optimization method should not simply search for any molecule with a higher predicted score; it should preferably propose local modifications around the original scaffold. Otherwise, the generated molecule may belong to a substantially different chemotype and become less relevant to the original lead compound.
>
> This is also the key motivation behind our method. There is an inherent trade-off in molecular optimization: if the model focuses only on preserving similarity, it may generate trivial edits with little property improvement; if it focuses only on property gain, it may drift far away from the source molecule. Our goal is to make this locality--gain trade-off explicit and controllable.
>
> Algorithmically, SCPT differs from standard methods in three main aspects.
>
> First, unlike standard supervised fine-tuning on isolated source--target molecule pairs, SCPT constructs preference triplets of the form
>
> $$
> \langle \text{scaffold},\ \text{better molecule},\ \text{worse molecule} \rangle .
> $$
>
> Here, “better” and “worse” are compared within the same scaffold-local neighborhood, so the model learns which local edit is preferable under a fixed scaffold condition.
>
> Second, unlike conventional black-box reward optimization, SCPT explicitly filters candidate pairs using property improvement, fingerprint similarity, local edit constraints, and common-substructure preservation before constructing preferences. Thus, the supervision is not only property-driven but also chemically localized.
>
> Third, by applying DPO on these scaffold-conditioned triplets, the model is aligned to prefer higher-gain local edits without treating molecular optimization as unconstrained generation.
>
> We agree with the reviewer that the current presentation can be improved. In the revised version, we will make the following changes:
>
> 1. We will add a clearer definition of “scaffold” in the Introduction and Section 3.1, together with an intuitive explanation of why scaffold preservation matters in lead optimization.
>
> 2. We will revise Section 3.2 to more explicitly contrast SCPT with standard pairwise SFT and reward-based molecular optimization.
>
> 3. We will add a concise comparison paragraph or table distinguishing prompt-only LLM generation, standard molecule-pair SFT, classical reward-based optimization, and our SCPT + SFT/DPO framework.
>
> 4. We will revise Figure 2 and its caption to label the three main stages more clearly: property-based filtering, similarity/local-edit filtering, and scaffold-conditioned triplet construction.
>
> Overall, we appreciate the reviewer’s suggestion. The revision will make the central contribution clearer: SCPT converts chemically motivated scaffold-preserving local-edit constraints into structured preference supervision, and then uses preference alignment to train an LLM editor that can navigate the trade-off between molecular property improvement and scaffold preservation.

---

> ### Comment · Action_Editor_BYmt · 2026-06-01
>
> Dear reviewer,
>
> The authors had responded your concerns a few days ago. Please check whether their response has addressed your concerns asap. Many thanks.
>
> Best,
>
> AE

---

> > ### Comment · Reviewer_14FM · 2026-06-01
> > **Revised Version**
> >
> > I think the authors response makes sense and would likely improve the paper significantly. I will read the revised version when it becomes available.

---

### Review · Reviewer_Xw8x · 2026-05-17

**Summary Of Contributions:**

This paper introduces Scaffold-Conditioned Preference Triplets (SCPT), a pipeline for aligning LLMs to perform scaffold-preserving molecular property optimization. The method constructs preference triplets ⟨scaffold, better, worse⟩ using similarity constraints, fragment-level edit filters, and property-margin requirements, then applies supervised fine-tuning followed by direct preference optimization to train conditional molecular editors.

Strengths:
- The manuscript is well motivated by pointing out a critical gap in molecular optimization: current methods struggle to balance property improvement with scaffold preservation. And the focus on lead-based optimization directly mirrors real drug discovery workflows.
- The pipeline is well-designed with multiple filters: ex.filters to ensure property improvements and enforce similarity threshold, single-fragment edit constraints for interpretability, etc
- Performance seems very impressive across benchmarks: i.e., near-saturated success rate (>95%) on single-property tasks with subtantial property gains while keeping similarity high
- Provides a fairly comprehensive comparision with general-purpose and chemistry focused LLMs.

Weaknesses:
- Lacks some experiment details: For the given dataset/corpus, what is the wall-clock training time for SFT and DPO stages? What about the computational infrastructure? They will help readers understand the scalability to larger models or datasets
- When does SCPT fail and what kind of errors does it make? Though success rates are proven to high, failure analysis would help practitioners know when the method is appropriate?
- Table 6 compares SCPT against non-LLM baselines (GraphGA, JTVAE, REINVENT) from 2018-2019. Notably absent are more recent molecular optimization methods, particularly diffusion-based approaches that have shown strong performance in the field. Including comparisons with more recent methods would more rigorously position SCPT's contribution.

More to discuss:
- Would the filtering process create data imbalance? For example, some scaffolds may have many triplets while others have few, which could affect the generality of the model.
- Realistic lead optimization may involve way more simultaneous constraints. Would SCPT pipeline still be scalable and relevant (given that there could be conflicting objectives such as potentcy vs solubility). RQ5 section is a good start with 3 property combinations.

**Audience:**

Yes

**Audience Explanation:**

Yes, the topic of improving molecular generation/optimization is relevant to TMLR's audience.

**Claims And Evidence:**

Yes

**Claims Explanation:**

- Table 1 and 2 supports the claims that SCPT pipeline provides trade off between scaffold preservation and property improvement
- Table 3 and 4 supports that preference alignment eliminates scaffold drift

**Requested Changes:**

In addition to "Weaknesses" above, here's a minor note:
- Figure 2 caption states the framework consists of “three main modules”. Should it be two (a and b)?

---

> ### Author Response · Authors · 2026-05-25
> **Response to Reviewer Comments on Experimental Details, Failure Analysis, Diffusion Baselines, and Limitations**
>
> We thank the reviewer for the constructive comments. We have revised the manuscript by adding training-cost details, a failure analysis, a recent diffusion baseline, and additional discussions on scaffold distribution and realistic multi-objective optimization.
>
> ### 1. Training time and computational cost
>
> We agree that the original manuscript lacked sufficient implementation details. We now report the wall-clock training time and hardware setup. We use LLaMA-3.1-8B and Mistral-2.0-Instruct-7B as backbones, with LoRA rank 8 and alpha 16. All training is conducted on a single RTX 4090 GPU with 24GB memory.
>
> | Stage | Epochs | Batch / Micro-batch | Time |
> |---:|---:|---:|---:|
> | SFT | 3 | 128 / 16 | ~45 min |
> | DPO |  3 | 128 / 8 | ~95 min |
>
> Thus, SCPT does not require full-parameter fine-tuning or large-scale compute, and can be reproduced with a single consumer-grade GPU.
>
> ### 2. Failure modes of SCPT
>
> We agree that success rate alone does not fully explain when SCPT fails. We therefore add a systematic failure analysis. We classify failed cases into Invalid, Wrong Direction, and Insufficient Improvement. Invalid means that the generated molecule cannot be parsed by RDKit or evaluated by the oracle; Wrong Direction means that the target property changes in the opposite direction; Insufficient Improvement means that the direction is correct but the gain does not reach the threshold.
>
> | Setting | Avg. SR | Main failure mode | Interpretation |
> |---|---:|---|---|
> | Single-property | 93.3% | Invalid SMILES | The model usually learns the right direction, but some local edits are invalid. |
> | Multi-property | 69.4% | Wrong Direction / Insufficient Improvement | Satisfying multiple directions and thresholds is harder. |
>
> This shows that SCPT is stable for single-property optimization, where failures mostly come from molecular validity. For multi-property tasks, the difficulty shifts to satisfying several objectives simultaneously, consistent with realistic lead optimization. We will include a more detailed task-level and model-level failure analysis in the revised manuscript.
>
> ### 3. Recent diffusion baseline
>
> We agree that Table 6 should include a more recent diffusion-based method. We therefore add GenMol, a 2025 diffusion-based molecular generalist, as an additional baseline. We also clarify the baseline-selection principle. Many diffusion models are mainly designed for de novo generation, 3D or structure-conditioned generation, or fragment generation, whereas our task is source-conditioned scaffold-preserving local editing. Strictly adapting these models would require adding source/scaffold conditioning and retraining with our oracle thresholds. Thus, we use GenMol as a recent executable diffusion comparison.
>
> GenMol shows strong validity and uniqueness, but limited scaffold retention and multi-property success in our setting.
>
> | GenMol setting | Budget 20 | Budget 200 | Observation |
> |---|---:|---:|---|
> | Single-property SR | 10.8%–62.8% | 34.2%–76.3% | SR improves with sampling budget. |
> | Single-property SIM | ~0.43–0.45 | ~0.36–0.39 | Similarity decreases with budget. |
> | Multi-property SR | 0%–2.5% | 0%–4.4% | Joint constraints remain difficult. |
>
> These results suggest that additional diffusion sampling improves property scores mainly through broader chemical-space exploration, rather than stable scaffold-preserving local editing. In contrast, SCPT focuses on controllable local optimization around a given source molecule.
>
> ### 4. Scaffold distribution and data imbalance
>
> We also clarify the difference between leakage control and scaffold imbalance. Test molecules are deliberately excluded from training to prevent train-test leakage and evaluate generalization; this is a stricter evaluation design, not a filtering artifact. The relevant issue is the training distribution after SCPT filtering: since valid triplets must satisfy property gain, similarity, local-edit, and validity constraints, different scaffolds may contribute different numbers of examples. We have added a scaffold-level analysis and discuss scaffold-balanced sampling, reweighting, and active data acquisition as future remedies.
>
> ### 5. Realistic multi-objective optimization
>
> We agree that real lead optimization may involve more simultaneous and conflicting constraints, such as potency, solubility, toxicity, BBBP, synthesizability, and selectivity. Our RQ5 is an initial study of whether single- and two-property supervision can generalize to unseen three-property combinations. We revise the discussion to state this limitation and suggest Pareto-aware preference construction, constraint-conditioned prompting, controllable objective weights, and active data construction as future directions.
>
> ### 6. Figure 2 caption
>
> We corrected the Figure 2 caption. The framework contains two main stages—SCPT data construction and preference-based alignment—rather than “three main modules.”

---

> > ### Comment · Reviewer_Xw8x · 2026-05-30
> >
> > I appreciate the authors’ prompt response. My concerns have been addressed.

---

### Review · Reviewer_SZig · 2026-06-13

**Summary Of Contributions:**

This paper proposes Scaffold-Conditioned Preference Triplets (SCPT), a data construction pipeline for scaffold-preserving molecular optimization with LLMs. The method builds `<scaffold, better, worse>` preference triplets using property improvement, similarity, and local-edit constraints, then applies SFT and DPO to train molecular editors. The paper shows that SCPT improves the trade-off between property optimization and scaffold preservation and provides empirical analysis of the Pareto relationship between similarity and relative improvement.

**Audience:**

Yes

**Audience Explanation:**

Molecular optimization is an active area in machine learning, and the use of preference-based alignment for controllable scientific generation should be relevant to readers interested in LLMs, molecular generation, and structured optimization.

**Claims And Evidence:**

Yes

**Claims Explanation:**

The paper includes sufficient experiments across single-property, multi-property, and compositional generalization settings. The ablation studies on similarity thresholds, property-gap thresholds, and DPO hyperparameters support the main claim that SCPT provides a controllable trade-off between scaffold preservation and property improvement.

**Requested Changes:**

The paper is generally clear and well supported. I only have a few minor requested changes.

1. The authors could add a short paragraph summarizing the practical takeaway from the similarity/property-improvement trade-off. For example, when should users choose a stricter similarity setting, and when is a more aggressive improvement-oriented setting preferable?

2. The paper would benefit from a clearer distinction between the role of SFT and DPO. The results suggest that SFT mainly learns feasible local edits, while DPO shifts the model toward higher-gain edits with some similarity cost. Stating this more explicitly would help readers interpret the experimental results.

3. Please clarify in the main text that the final optimized molecule is selected from multiple generated candidates using the property oracle. This detail is important for interpreting SR and RI.

---

> ### Author Response · Authors · 2026-06-14
> **Response to Requested Clarifications**
>
> We thank the reviewer for these helpful suggestions. We agree that these points can be clarified more explicitly in the main text, and we will revise the manuscript accordingly.
>
> First, we will add a short paragraph discussing the practical implication of the similarity/property-improvement trade-off. Specifically, we will clarify that stricter similarity settings are preferable when scaffold preservation and conservative lead refinement are the main goals, while more aggressive settings are more suitable when larger property gains are prioritized and moderate scaffold changes are acceptable. This will help readers understand how to choose an operating point along the observed Pareto frontier.
>
> Second, we agree that the distinction between SFT and DPO should be stated more clearly. In the revised manuscript, we will emphasize that SFT mainly teaches the model to perform valid and feasible local molecular edits under the given instruction format, whereas DPO further shifts the model toward edits with larger property gains based on scaffold-conditioned preferences. We will also explicitly note that this usually comes with a moderate decrease in similarity, which is consistent with the trade-off observed in the experiments.
>
> Third, we will clarify the inference and evaluation protocol in the main text. In particular, we will state that for each source molecule, the model generates multiple candidate molecules, and the final optimized molecule is selected according to the task-specific property oracle before computing SR and RI. We agree that this detail is important for interpreting the reported optimization performance.

---

### Comment · Reviewer_14FM · 2026-06-28
**Author Names**

The names of the authors are visible in the revised manuscript. I was under the impression that TMLR submissions were supposed to be double blind.

I do not know any of the authors and have submitted my recommendation as if I did not see their names. I leave it to the editor to decide what actions to take

---

> ### Author Response · Authors · 2026-06-29
>
> Thank you for pointing this out. We apologize for the oversight: in preparing the revised manuscript after the first review round, we inadvertently uploaded a version containing author-identifying information.
>
> We have now uploaded a corrected anonymized revised manuscript, and we have checked the manuscript and supplementary materials for other identifying information.
>
> We leave any further procedural decision to the Action Editor.